# Mechanical coupling coordinates microtubule growth

Bonnibelle K Leeds[1], Katelyn F Kostello[1], Yuna Y Liu[1], Christian R Nelson[2], Sue Biggins[2], Charles L Asbury[1]*

[1]Department of Physiology & Biophysics, University of Washington, Seattle, United States; [2]Basic Sciences Division, Fred Hutchinson Cancer Research Center, Seattle, United States

*For correspondence:
casbury@uw.edu

Competing interest: The authors declare that no competing interests exist.

**Abstract** During mitosis, kinetochore-attached microtubules form bundles (k-fibers) in which many filaments grow and shorten in near-perfect unison to align and segregate each chromosome. However, individual microtubules grow at intrinsically variable rates, which must be tightly regulated for a k-fiber to behave as a single unit. This exquisite coordination might be achieved biochemically, via selective binding of polymerases and depolymerases, or mechanically, because k-fiber microtubules are coupled through a shared load that influences their growth. Here, we use a novel dual laser trap assay to show that microtubule pairs growing *in vitro* are coordinated by mechanical coupling. Kinetic analyses show that microtubule growth is interrupted by stochastic, force-dependent pauses and indicate persistent heterogeneity in growth speed during non-pauses. A simple model incorporating both force-dependent pausing and persistent growth speed heterogeneity explains the measured coordination of microtubule pairs without any free fit parameters. Our findings illustrate how microtubule growth may be synchronized during mitosis and provide a basis for modeling k-fiber bundles with three or more microtubules, as found in many eukaryotes.

## eLife assessment

In this technically advanced and **important** piece of work, the authors study the coordination of microtubule growth in kinetochore fibers using force spectroscopy and numerical simulations. With **compelling** evidence the authors address the question of how microtubules, which naturally exhibit variable growth rates, can coordinate their behavior by mechanical coupling so as to function as a single unit in generating forces during chromosome segregation.

## Introduction

In the mitotic spindles of many eukaryotes, each chromosome is attached to a bundle of microtubules (*Kiewisz et al., 2022*; *McDonald et al., 1992*; *O'Toole et al., 2020*; *VandenBeldt et al., 2006*) that grow and shorten in unison, acting as a single unit to produce the forces necessary to move the chromosome (*Cimini et al., 2004*). Microtubules are microns-long protein polymers composed of αβ-tubulin dimers with their 'minus' ends oriented toward the spindle poles and their more dynamic 'plus' ends oriented toward the chromosomes. Microtubule plus-ends, or tips, attach to chromosomes via protein complexes called kinetochores to form 'k-fiber' bundles. Kinetochore-attached tips within a single k-fiber remain very close to one another (within ~0.2 μm along the bundle axis) (*O'Toole et al., 2020*) even as the bundle grows and shortens (*Cimini et al., 2004*) by the addition and loss of tubulin subunits at each tip (*Maddox et al., 2003*). The number of microtubules per k-fiber varies across species; human RPE1 cells have 10–15 microtubules per k-fiber, while marsupial PTK2 cells have more than 20, and cervid Indian Muntjac cells have up to 60 (*O'Toole et al., 2020*; *Drpic*

**Figure 1.** Intrinsic variability in microtubule growth could be coordinated by mechanical coupling. (**a**) Microtubules (MTs) grow at intrinsically variable rates that increase with force, on average. Example recordings show plus-end position versus time for individual growing microtubules subject to a constant tensile force of either 2 or 6 pN (thin green or magenta traces, respectively). At each force, 10–14 example recordings are shown, together with the mean position versus time (thick traces, computed from $N$=25 or $N$=10 individual recordings at 2 or 6 pN, respectively). Individual traces are from *Akiyoshi et al., 2010*. (**b**) Mechanism by which mechanical coupling could coordinate growing microtubules. In this example, two microtubules share a total force, $F_{TOT}$, through two spring-like connections to their plus-ends. If one microtubule stochastically lags behind, it will experience more tension than the leading microtubule due to differential stretching of their spring-connections. Because tension tends to accelerate plus-end growth (as illustrated in (**a**)), the higher tension will tend to accelerate the growth of the lagging microtubule, causing it to catch up.

*et al., 2018*). Budding yeast lack k-fiber bundles, having just a single microtubule per kinetochore (*Winey et al., 1995*), although microtubules attached to sister kinetochores on opposite sides of the yeast spindle must still coordinate with each other. Uncovering how microtubule tips maintain a high degree of synchrony within k-fibers of up to 60 microtubules is critical to understanding how chromosomes are accurately segregated during normal mitosis and why even modest perturbations of kinetochore-attached microtubule dynamics can lead to mis-segregation (*Bakhoum and Compton, 2012*; *Bakhoum et al., 2009*; *Ertych et al., 2014*; *Stumpff et al., 2014*).

The tightly coordinated growth of microtubules within a k-fiber bundle is remarkable because individually, microtubule tips exhibit highly variable growth speeds and switch stochastically between distinct states of growth and shortening (*Brouhard, 2015*). Microtubule growth speed varies considerably in purified tubulin systems (*Figure 1a*; *Cleary et al., 2022*; *Gildersleeve et al., 1992*; *Mahserejian et al., 2022*), in cell extracts (*Geisterfer et al., 2020*), and in live cells (*Cassimeris et al., 1988*; *Gaudenz et al., 2010*; *Yamashita et al., 2015*). Part of this heterogeneity can be attributed to pausing, when microtubules stochastically stop and start growing during periods of net growth (*Amoah-Darko and White, 2022*; *Mahserejian et al., 2022*). How the intrinsic heterogeneity of microtubule growth is coordinated to maintain sub-micrometer proximity between k-fiber tips remains a mystery.

One possible mechanism that might coordinate k-fiber microtubules is selective binding of microtubule-regulating proteins (*Maiato et al., 2003b*; *McAinsh and Meraldi, 2011*; *Tirnauer et al., 2002*). Many microtubule-associated proteins that are known to regulate growth *in vitro* are implicated in modulating k-fiber dynamics and kinetochore movements in cells, including TOG-family proteins such as XMAP215 (*Brouhard et al., 2008*) and CLASP (*Al-Bassam et al., 2010*; *Maiato et al., 2003a*; *Sousa et al., 2007*), mitotic kinesins such as MCAK and Kif18A (*Sanhaji et al., 2011*; *Stumpff et al., 2012*), and other plus-end-trackers such as EB1 (reviewed in *Amin et al., 2019*). The localization of some of these regulators is enriched near kinetochores specifically when a kinetochore is moving anti-poleward and its k-fiber is growing longer (*Amaro et al., 2010*; *Castrogiovanni et al., 2022*; *Tirnauer et al., 2002*). However, whether such selective enrichment is sufficient to explain the very tight coordination across many individual microtubule tips within a k-fiber is unclear.

Another, simpler mechanism is suggested by the fact that microtubule dynamics are also strongly affected by mechanical force. Tension applied to individual microtubules *in vitro* biases them toward growth (*Figure 1a*; *Akiyoshi et al., 2010*), while compression biases them toward shortening (*Dogterom and Yurke, 1997*; *Janson et al., 2003*). Theoretically, these mechanical effects should promote synchronization of any group of microtubules whose tips, like the tips of kinetochore-attached microtubules within a k-fiber, are all coupled to the same load (*Banigan et al., 2015*; *Schwietert and Kierfeld, 2020*; *Skibbens et al., 1993*). To illustrate how, consider a hypothetical k-fiber composed of just two microtubules sharing a total load, $F_{TOT}$. The load is transmitted to the microtubule tips through the kinetochore, which has elasticity (*Cojoc et al., 2016*; *Renda et al., 2020*) that we represent as spring-like connections from the kinetochore to each tip (*Figure 1b*). Initially, the two microtubules might have equal lengths and share the load equally, each one carrying half of $F_{TOT}$ (*Figure 1b*, step 1). If one tip happens to grow more slowly than the other, then its spring will be stretched more, and it will start carrying more tension than the other tip (*Figure 1b*, step 2). Conversely, the faster-growing tip will start carrying less tension. Because tension accelerates microtubule growth, the lagging tip will tend to accelerate, while the leading tip will tend to decelerate, such that, on average, they return to having equal lengths and tensions (*Figure 1b*, step 3).

Mechanical coupling was proposed decades ago as an explanation for coordination between sister k-fibers (*Gardner et al., 2005*; *Khodjakov et al., 1996*; *Skibbens et al., 1995*; *Skibbens et al., 1993*; *Wan et al., 2012*), but how effectively it can coordinate microtubules within a k-fiber has not been determined. Compressive forces can cause a bundle of microtubules growing *in vitro* to switch cooperatively into a shortening state (*Laan et al., 2008*), but k-fiber bundles are usually under tension, not compression (*Khodjakov and Rieder, 1996*; *Waters et al., 1996*). Theoretical studies have explored the effects of shared tensile and compressive loads on cooperative switching and bi-stability in microtubule bundles (*Banigan et al., 2015*; *Das et al., 2014*; *Ghanti and Chowdhury, 2015*; *Schwietert and Kierfeld, 2020*; *Zelinski and Kierfeld, 2013*). However, these studies have generally neglected any differences in growth speed between microtubules and therefore cannot address how much variability exists, nor how such variability is suppressed. Furthermore, to our knowledge, the possibility that part of this variability (and suppression thereof) can be attributed to tension-dependent pausing during microtubule growth has not been explored.

Here, we use a novel dual laser trap assay to test directly how well a constant shared load can coordinate a pair of microtubules growing *in vitro*. We show that when microtubule pairs are mechanically coupled to a single load through a material of sufficient stiffness, their growth is tightly coordinated. By re-analyzing recordings of individual microtubules growing under constant force (from *Akiyoshi et al., 2010*) and examining their stochastic pausing kinetics, we show that pausing is suppressed by tension. We then develop a theoretical model, based on the measured behaviors of individual growing microtubules, to describe both the natural variability and force dependence of growth using no free fit parameters. Simulations based on this model fit the dual-trap data well. Altogether, this work illustrates how mechanical coupling may coordinate microtubule growth and provides a basis for modeling k-fibers of three or more microtubules under a shared load.

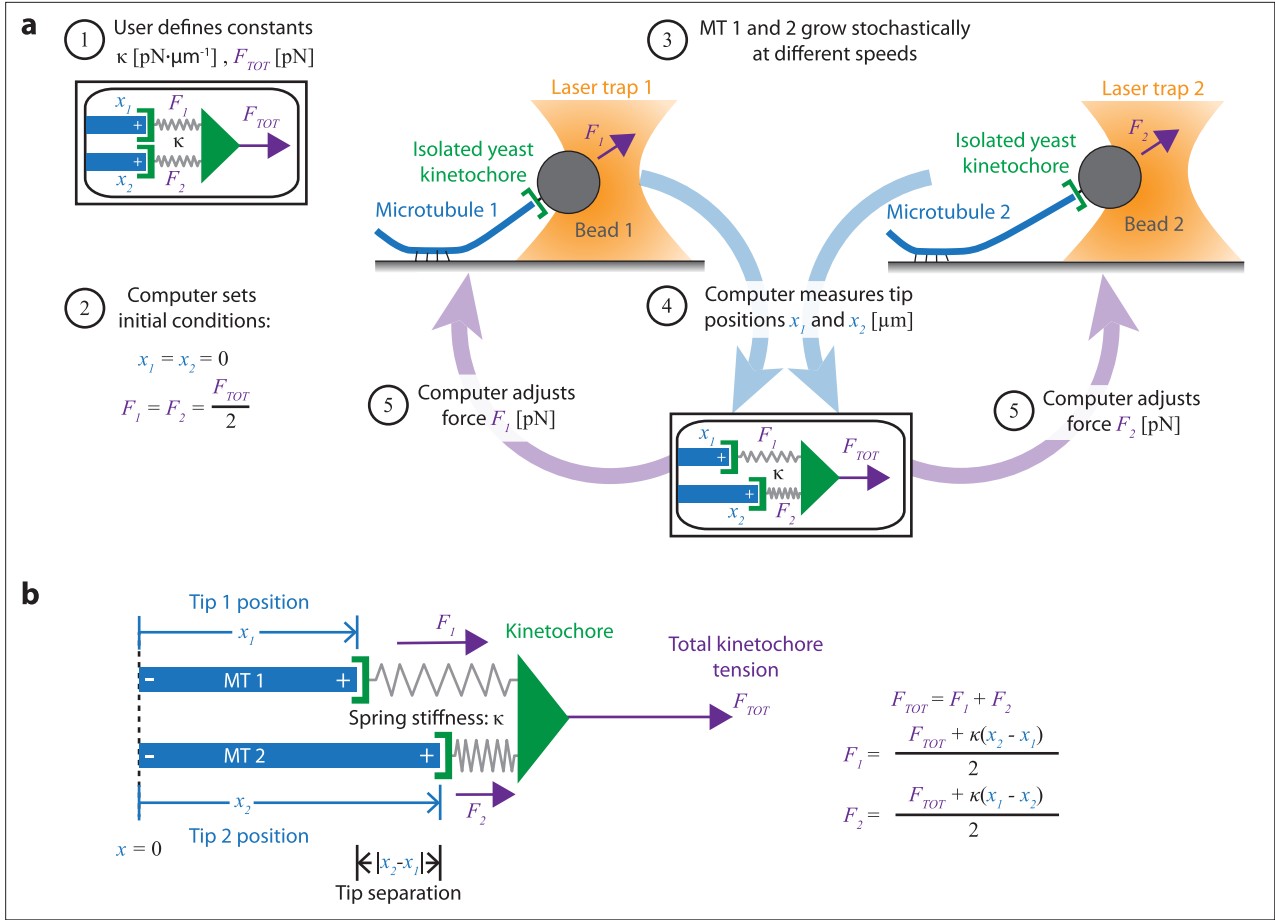

**Figure 2.** Dual-trap assay to measure the coordination of two real microtubules *in vitro* coupled together by simulated spring-like connections to a shared tensile load. (**a**) Schematic of the dual-trap assay, which measures the effects of the spring-coupler model (detailed in (b)) on real microtubules (MTs) growing *in vitro*. Two dynamic microtubules are grown from seeds anchored to two separate coverslips on two separate stages, each with their own microscope and laser trap. Each laser trap applies tension to its respective microtubule plus-end via a bead decorated with isolated yeast kinetochores, and each tension is adjusted dynamically under feedback control by a shared computer. Initially (steps 1 and 2), microtubule plus-ends are considered to have the same position ($x_1=x_2=0$) so that each bears half of the total user-input tension ($F_{TOT}$). Over time, as microtubules grow at different rates (step 3), the computer dynamically adjusts the tension on each microtubule according to the spring-coupler model (steps 4 and 5), so that the higher the tip separation ($|x_1 - x_2|$) and stiffness ($\kappa$), the larger the difference in tension between the two microtubules. (**b**) Spring-coupler model used to represent the physical connection between two microtubule plus-ends via a kinetochore. Both springs have identical stiffnesses ($\kappa$). The kinetochore is assumed to be in force equilibrium such that the sum of the spring forces is equal to the total external force, $F_{TOT}$, which is kept constant in the current study.

## Results

### Dual-trap assay to measure coordination between paired microtubules *in vitro*

To study how mechanical coupling affects microtubule growth, we developed a novel technique which we call the dual-trap assay (*Figure 2*). The dual-trap assay is based on our previously developed single-trap (force-clamp) assay (*Akiyoshi et al., 2010*; *Miller et al., 2016*; *Sarangapani et al., 2013*), in which dynamic microtubules are grown from stabilized seeds bound to a biotinylated coverslip. Using the single laser trap, we attach an individual bead decorated with isolated yeast kinetochores to the growing plus-end of a single microtubule. A computer then continuously measures the bead position and adjusts the trap to exert a precise, constant level of tension on the microtubule via the kinetochore-decorated bead. Under this persistent feedback-controlled tension, kinetochore-beads typically track with the microtubule tips even as the tips stochastically grow and shorten.

Our new dual-trap assay uses two separate laser trapping microscopes, located adjacent to one another in the same room and connected to a single computer. On each of the two instruments,

we attach a kinetochore-decorated bead to a dynamic microtubule plus-end. The computer then simultaneously monitors and controls the forces on both microtubules. Rather than keeping the force constant on each microtubule, the computer adjusts the forces dynamically to simulate an elastic coupling of both plus-ends to a single shared load (*Figure 2a*). Thus far, we have simulated only purely elastic couplers, where both coupling springs are linear (Hookean) with stiffness, $\kappa$ (*Figure 2b*). In the future, more complex coupling materials with viscoelasticity (*Cojoc et al., 2016*) or strain-stiffening (*Schwietert et al., 2022*; *Volkov et al., 2018*) could also be simulated.

To begin a dual-trap experiment, we first choose the spring stiffness, $\kappa$, and the total shared load, $F_{TOT}$, which are kept constant (*Figure 2a*, step 1). After a kinetochore-decorated bead is attached to a growing plus-end on each of the two instruments, feedback-control is initiated and the two plus-ends are arbitrarily considered to be parallel, with tips side-by-side (i.e. both at $x_1=x_2=0$) and sharing the load equally ($F_1=F_2 = \frac{1}{2} \cdot F_{TOT}$) (*Figure 2a*, step 2). Because microtubule growth is intrinsically variable, the two microtubules subsequently grow at different speeds (*Figure 2a*, step 3). The computer then dynamically monitors the bead positions (*Figure 2a*, step 4) and adjusts the forces, $F_1$ and $F_2$ (*Figure 2a*, step 5), according to the elastic coupling model (*Figure 2b*). For purely elastic couplers, the force difference across the two microtubules equals the tip separation, $x_2 - x_1$), multiplied by the coupling stiffness (, $\kappa$. When one microtubule grows more quickly than the other, tension on the leading (faster-growing) microtubule decreases and tension on the lagging (slower-growing) microtubule increases to maintain a constant total force on the pair, $F_{TOT}$. Forces in our dual-trap feedback system are updated at a rate of 50 Hz, which is more than sufficient for accurately simulating the elastic coupling between two microtubules (*Figure 3—figure supplement 1*). For all the experiments described below, $F_{TOT}$ = 8 pN. Thus, each microtubule generally experiences between 1 and 7 pN of tension, close to the estimated average of 4–6 pN experienced by kinetochore-microtubules in budding yeast (*Chacón et al., 2014*; *Parmar et al., 2023*).

## Mechanical coupling coordinates microtubule pairs *in vitro*

To measure the effects of mechanical coupling, we recorded the growth of many pairs of microtubules coupled to a constant shared load ($F_{TOT}$ = 8 pN) either through relatively soft springs, with $\kappa$=1 pN·µm⁻¹, or through springs with five-fold greater stiffness, $\kappa$=5 pN·µm⁻¹. These two stiffnesses span a range near the minimum elasticities estimated for mitotic chromosomes (*Nicklas, 1983*; *Poirier et al., 2002*), for pericentromeres (*Chacón et al., 2014*), and for elements of the kinetochore (*Schwietert et al., 2022*; *Volkov et al., 2018*). When coupled through soft springs ($\kappa$=1 pN·µm⁻¹), microtubule pairs tended to grow at different rates and their tips often became separated by more than a micrometer (*Figure 3a*, left, and *Figure 3—figure supplement 1*, left). After 300 s, the average tip separation for soft-coupled pairs grew to 0.8±0.2 µm (mean ± SEM from *N*=10 pairs) (*Figure 3b*, left). When coupled through stiff springs ($\kappa$=5 pN·µm⁻¹), growth speeds were better matched, and the microtubule tips remained closer together (*Figure 3a*, right, and *Figure 3—figure supplement 1*, right). The average tip separation for stiff-coupled pairs after 300 s was 0.25±0.06 µm (mean ± SEM from *N*=10 pairs) (*Figure 3b*, right), significantly lower than for soft-coupled pairs (p=0.03 based on a Wilcoxon-Mann-Whitney two sample rank test).

Because the plus-ends of microtubules within k-fiber bundles are rarely separated by more than 0.8 µm from each other *in vivo* (*O'Toole et al., 2020*), we used Kaplan Meier survival analysis to ask how often the pairs of coupled microtubule tips in our experiments exceeded this threshold separation. Consistent with their lower average tip separation, stiff-coupled microtubule pairs were much less likely to exceed a separation of 0.8 µm than their soft-coupled counterparts (p=0.01, log-rank test; *Figure 3c*), which sometimes separated by 0.8 µm in as little as 60 s. Even after 400 s, more than 80% of the stiff-coupled pairs had tip separations that remained below 0.8 µm, while fewer than half of the soft-coupled pairs had tip separations below 0.8 µm. (Similar survival analyses using thresholds of 0.4, 0.6, and 1.2 µm are provided in *Figure 6—figure supplement 2*).

Even at shorter time periods, the average behavior of soft- and stiff-coupled microtubule pairs was distinct. During the first 100 s of growth, the average rate of tip separation for soft-coupled pairs was 0.23±0.04 µm·min⁻¹ (mean ± SEM from *N*=23 pairs), while the average rate of tip separation for stiff-coupled pairs was only 0.14±0.02 µm·min⁻¹ (mean ± SEM from *N*=26 pairs; p=0.03 based on a Wilcoxon-Mann-Whitney two sample rank test). Notably, the average rate of tip separation for stiff-coupled microtubule pairs decreased as their average tip separation increased, suggesting they were

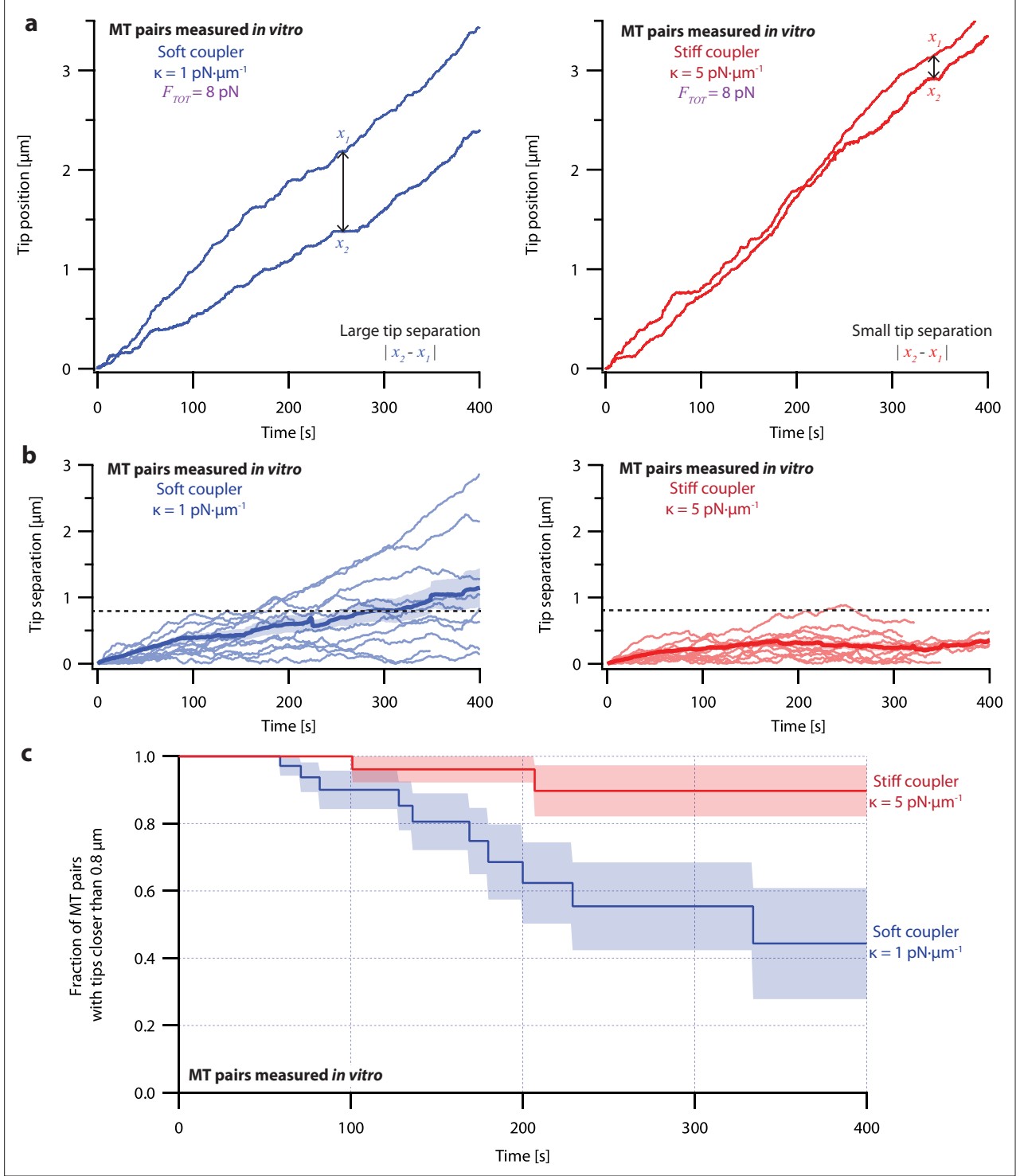

**Figure 3.** Microtubule pairs growing *in vitro* are coordinated by mechanical coupling. (**a**) Example dual-trap assay recordings showing real microtubule (MT) plus-end positions over time for representative microtubule pairs coupled via soft (1 pN·µm⁻¹, left) or stiff (5 pN·µm⁻¹, right) simulated spring-couplers. All dual-trap data was recorded with $F_{TOT}$ = 8 pN. (**b**) Tip separation ($|x_1 - x_2|$) between microtubule plus-end pairs that were coupled with either soft (left) or stiff (right) simulated spring-couplers over 400 s of growth. Light colors show tip separations for 10 individual microtubule pairs with each coupler stiffness, while dark colors show mean tip separation for all recorded pairs (N=50 and N=43 recordings with soft and stiff couplers, respectively). (**c**) Fraction of microtubule pairs whose plus-ends remained within 0.8 µm of each other over time. (In other words, the fraction of pairs at any given time whose tip separation had not exceeded the dashed line in (**b**).) Shaded regions in (**b**) and (**c**) show SEMs from N=50 and N=43 recordings with soft and stiff couplers, respectively.

*Figure 3 continued on next page*

*Figure 3 continued*

The online version of this article includes the following figure supplement(s) for figure 3:

**Figure supplement 1.** Additional dual-trap assay recordings showing tip positions and forces on pairs of growing microtubules.

approaching a steady-state level of separation (*Figure 3b*, right). In contrast, the average rate of tip separation for soft-coupled microtubule pairs remained approximately constant throughout the 400 s measurement, suggesting that soft-coupled pairs would require more time to approach a steady state. Altogether, these observations show that the growth of microtubule pairs can be tightly coordinated, even over relatively short (<100 s) timescales, by mechanically coupling their tips to a shared load through couplers of sufficient stiffness.

## Models of microtubule growth variability and force sensitivity can be derived from recordings of individual microtubules

Coordination of microtubule growth can be thought of as the suppression of variability in microtubule growth speed. Understanding how mechanical coupling suppresses this variability requires a realistic model of how the growth of individual microtubules varies over time and between microtubules, as well as how growth behaviors are influenced by force. Previous simulations of microtubule bundles (*Banigan et al., 2015*; *Schwietert and Kierfeld, 2020*), based on force-clamp measurements of individual microtubules (*Akiyoshi et al., 2010*), have suggested that mechanical coupling might coordinate the stochastic switching of microtubules between shortening and growth, such that sister k-fibers are nearly always kept in opposite states. However, these models assumed identical growth speeds for all microtubules under the same force, and therefore cannot address how mechanical coupling might coordinate differences in growth speed. We sought to create a model of microtubule growth that explains both the variability in microtubule growth speeds and how that variability is suppressed by sufficiently stiff mechanical coupling, as observed in the dual-trap assay.

To characterize the variability and force dependence of microtubule growth, we re-examined the force-clamp recordings by *Akiyoshi et al., 2010*. This large dataset (*N*=356 recordings) was recorded with an identical setup to our dual-trap experiments, except that each microtubule was grown individually with a single laser trap exerting a constant level of tension (between 0.5 and 18 pN). Using this dataset, we created two models derived from the natural variability of individual kinetochore-attached microtubules and their average behaviors as a function of force. Below, we begin by briefly describing a simple non-pausing model, which was developed first and served as a foundation for our second, more successful pausing model. The pausing model reasonably explains our dual-trap results by incorporating both stochastic pausing and intrinsic differences in microtubule growth speed.

## A simple non-pausing model predicts larger tip separations than measured *in vitro*

When averaged across time and across many individual microtubules, mean growth speed increases exponentially with applied tension (*Akiyoshi et al., 2010*), such that

$$v_g = v_{g0} * exp\left(\frac{F}{F_g}\right) \tag{1}$$

where $v_{g0}$ represents the unloaded growth speed, $F$ represents the applied tension, and $F_g$ represents the sensitivity with which the growth speed increases with tension. However, individual microtubules show considerable variability around this mean behavior, with some microtubules remaining persistently slower than others for many minutes (*Figure 1a*). In our first model, we adopted the simple assumptions that this variability stems entirely from intrinsic differences in unloaded microtubule growth speed, $v_{g0}$, and that all microtubules have identical force-sensitivity (with $F_g$ = 8.4 pN, based on the exponential curve that best fit the mean growth speed-vs-force data *Akiyoshi et al., 2010*). In principle, such static heterogeneity in growth speed might arise from persistent differences in microtubule tip structure (*Cleary and Hancock, 2021*; *Mahserejian et al., 2022*; *Rai et al., 2021*) or from variation in the numbers of protofilaments per microtubule (*Chrétien et al., 1992*; *Rai et al., 2021*).

For each simulated microtubule, we randomly chose an individual growth speed measurement from the force-clamp dataset (*Akiyoshi et al., 2010*), then extrapolated the microtubule's intrinsic unloaded speed ($v_{g0}$) from the chosen point ($v_g, F$) using equation [1] with $F_g$ = 8.4 pN. The growth speed of this simulated microtubule then followed the extrapolated curve as a function of force. To simulate a pair of coupled microtubules, we made two such random draws from the dataset, defining two distinct exponential ($v_g$ -versus-$F$) relations for the two microtubules. (Bundles of three or more microtubules could be simulated in the same manner by making additional draws.) We used a simple Monte Carlo-based method (*Gardner and Odde, 2010*) to simulate traces of tip position versus time for the pair. As in our dual-trap experiments, the simulated tip pairs were coupled with either soft or stiff elastic springs to a constant shared load, $F_{TOT}$ = 8 pN. Both the variability and force-dependence of growth in the model were derived directly from the experimental force-clamp data, with no adjustable ('free') parameters.

Despite its direct derivation from force-clamp data, the non-pausing model could not explain the behavior of microtubule pairs measured in the dual-trap assay (*Figure 4*). Simulated microtubule pairs developed much larger tip separations than real microtubule pairs over time, implying that this model overestimates real microtubule variability. The simulated microtubule pairs showed especially poor agreement with real microtubule pairs under the stiff-coupled condition (*Figure 4c*, red curves), indicating that the non-pausing model also does not capture the degree to which real microtubules are coordinated by mechanical coupling.

## Microtubule growth is interrupted by stochastic, force-dependent pauses

Our simple non-pausing model lacked a notable behavior exhibited by microtubules that could influence their response to tension: Microtubule growth is frequently interrupted by stochastic pauses of varying durations, both in cells (*Gaudenz et al., 2010*; *Sousa et al., 2007*) and in purified tubulin systems (*Mahserejian et al., 2022*; *Stumpff et al., 2012*). Indeed, pauses recorded during our force-clamp (*Figure 1a* and *Figure 5—figure supplement 1b*) and dual-trap assays (*Figure 3a* and *Figure 3—figure supplement 1*) were frequent and often lasted tens of seconds. Presumably, these pauses occur because the configuration of the growing microtubule tip fluctuates and sometimes becomes transiently refractory to the incorporation of new tubulin subunits (*Cleary and Hancock, 2021*; *Mahserejian et al., 2022*). We hypothesized that entry and exit into the pause state might be force-dependent processes, and that pausing might therefore be a key aspect of microtubule force-sensitivity.

To test whether pause entry and exit rates vary with force, we reanalyzed the force-clamp dataset (*Akiyoshi et al., 2010*), applying a 2 s sliding window to identify intervals during each recording (*N*=356) when the instantaneous growth speed fell to near-zero, indicative of a pause (*Figure 5—figure supplement 1*; see Methods for details about our pause marking procedure). The intervals in-between these pauses were designated as 'runs.' Distributions of run and pause durations were approximately exponential (*Figure 5—figure supplement 2*), consistent with a relatively simple two-state kinetic model of microtubule growth (*Figure 5—figure supplement 1a*), in which pause entry and exit are considered Poisson processes. For each force level, we estimated the pause entry and exit rates, $k_{EN}$ and $k_{EX}$, by counting the total number of observed entrances into and exits out of pause and dividing by the total time spent in run and pause, respectively (*Figure 5—figure supplement 1a*).

This analysis revealed that tension delays pause entry and accelerates pause exit (*Figure 5a*). Both effects tend to decrease the time spent in the pause state for growing microtubule tips under high tension. In addition, the average growth speed during runs was accelerated by tension (*Figure 5b*). All three rates, pause entry, pause exit, and average growth speed during runs, were well-fit by single exponential functions of force (*Figure 5*; see *Table 1* for fit parameters). These observations show that stochastic pausing during microtubule growth is force-dependent and imply that pausing kinetics will strongly influence the response of growing microtubules to mechanical coupling.

## A pausing model can explain the variability between paired microtubule and their coordination by mechanical coupling

To examine how force-dependent pausing might affect the behavior of coupled microtubules, we created a second model in which growing microtubules switch stochastically between states of pause

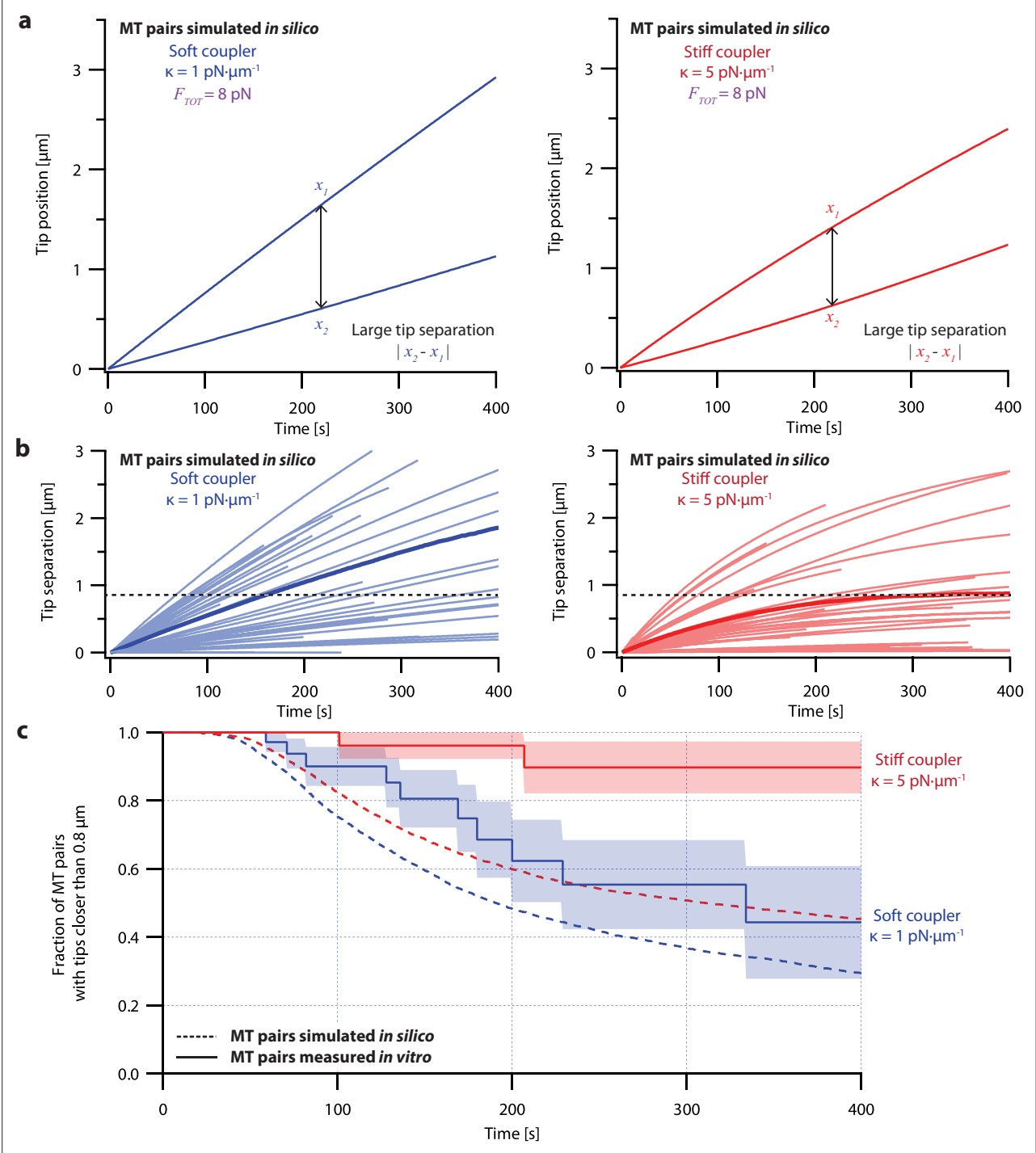

**Figure 4.** Simple non-pausing simulations fail to recapitulate the coordinated growth of mechanically coupled microtubule pairs. (**a**) Example simulations of microtubule (MT) plus-end positions over time using the non-pausing model. Simulated microtubule pairs were coupled via soft or stiff spring-couplers and shared a total load $F_{TOT}$ = 8 pN to match dual-trap assay conditions. (**b**) Tip separations between microtubule plus-end pairs that were simulated using the non-pausing model and coupled with either soft (left) or stiff (right) spring-couplers. Light colors show tip separations for $N$=40 individual simulated microtubule pairs, while dark colors show mean tip separation for $N$=10,000 simulated pairs. (**c**) Comparison between the fraction of *in silico* (simulated, dashed curves) and *in vitro* (real, solid curves) microtubule pairs whose plus-ends remained within 0.8 μm of each other over time. Shaded regions show SEMs from $N$=50 and $N$=43 *in vitro* recordings with soft and stiff couplers, respectively. Soft- and stiff-coupled microtubule pairs were each simulated $N$=10,000 times over 400 s of growth in both (**b**) and (**c**). Simulation parameters for (**a**), (**b**), and (**c**) can be found in *Table 1*.

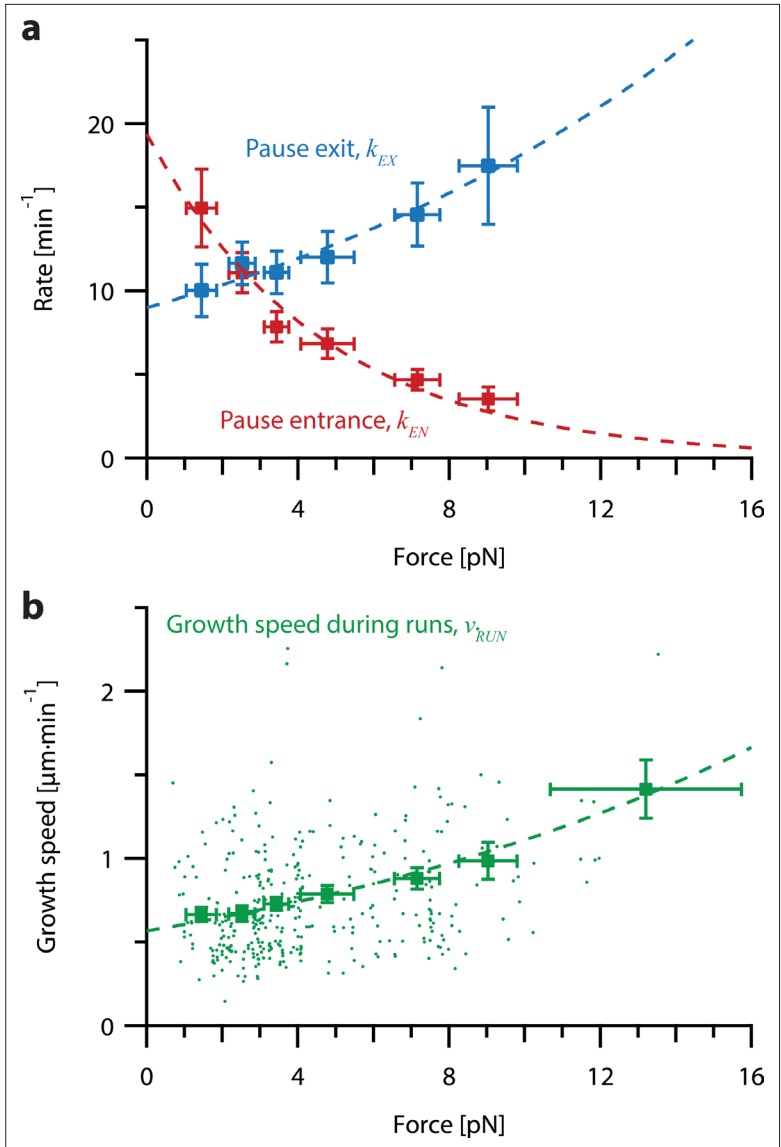

**Figure 5.** Increasing tension on individual growing microtubules suppresses pausing and accelerates growth during runs. (**a**) As tension increases on a growing microtubule tip, the rate of pause entrance decreases, while the rate of pause exit increases. Mean pause entrance and exit rates ($k_{EN}$ and $k_{EX}$) were estimated as described in *Figure 5—figure supplement 1* and are plotted as functions of force. Error bars show counting uncertainties from $N$=25–85 individual microtubule recordings and dashed curves show least squares exponential fits. (**b**) As tension on a microtubule tip increases, the growth during runs (i.e. in-between pauses) is accelerated. Points show run speeds measured from recordings of individual microtubules subject to constant tensile forces (from *Akiyoshi et al., 2010*). Symbols represent mean run speeds ($v_{RUN}$) grouped according to the applied tensile force. Error bars show SEMs from $N$=9–85 individual microtubule recordings and dashed curve shows the least squares exponential fit. See *Figure 5—figure supplement 1* and Methods for details and *Table 1* for fit parameters.

The online version of this article includes the following figure supplement(s) for figure 5:

**Figure supplement 1.** Individual microtubules frequently pause during periods of overall growth.

**Figure supplement 2.** Pause and run durations are exponentially distributed.

___

and run. Pause entry and exit were considered Poisson processes with kinetic rates dictated by the exponential fits of $k_{EX}$ and $k_{EN}$ versus force estimated from the force-clamp dataset (*Figure 5a*, dashed curves; *Table 1*). Intrinsic heterogeneity among microtubules was incorporated by randomly choosing an individual run speed measurement for each simulated microtubule from the force-clamp dataset (i.e. by randomly choosing one point in *Figure 5b*). We then extrapolated the microtubule's

**Table 1.** Parameter table for coupled microtubule pairs simulated for non-pausing and pausing models.

Relevant parameters measured from constant force recordings from **Akiyoshi et al., 2010**. These parameters were used to simulate coupled microtubule pairs using each of the models described in the main text. The equation in which each parameter is used, as well as which models of microtubule growth use that parameter, are indicated.

| Parameter name | Parameter symbol | Relevant equation | Parameter estimate | Applicable models | Reference |
|---|---|---|---|---|---|
| Growth speed force sensitivity | $F_g$ | $$v_g = v_{g0} \exp\left(\frac{F}{F_g}\right)$$ | 8.4 pN | non-pausing | **Akiyoshi et al., 2010**, **Figure 4e** |
| Unloaded pause entrance rate | $k_{EN0}$ | $$k_{EN} = k_{EN0} \exp\left(\frac{F}{F_{EN}}\right)$$ | 19.4 min⁻¹ | pausing | **Figure 5a**, red |
| Pause entrance rate force sensitivity | $F_{EN}$ | $$k_{EN} = k_{EN0} \exp\left(\frac{F}{F_{EN}}\right)$$ | –4.6 pN | pausing | **Figure 5a**, red |
| Unloaded pause exit rate | $k_{EX0}$ | $$k_{EX} = k_{EX0} \exp\left(\frac{F}{F_{EX}}\right)$$ | 9.0 min⁻¹ | pausing | **Figure 5a**, blue |
| Pause exit rate force sensitivity | $F_{EX}$ | $$k_{EX} = k_{EX0} \exp\left(\frac{F}{F_{EX}}\right)$$ | 14.1 pN | pausing | **Figure 5a**, blue |
| Unloaded run speed | $v_0$ | $$v_{RUN} = v_0 \exp\left(\frac{F}{F_{RUN}}\right)$$ | 0.52 µm·min⁻¹ | neither | **Figure 5b** |
| Run speed force sensitivity | $F_{RUN}$ | $$v_{RUN} = v_0 \exp\left(\frac{F}{F_{RUN}}\right)$$ | 14.8 pN | pausing | **Figure 5b** |

intrinsic unloaded run speed ($v_0$) from the chosen point ($v_{RUN}, F$) by assuming an exponential force-dependence with sensitivity $F_{RUN}$ = 14.8 pN (based on the dashed curve in **Figure 5b**; **Table 1**). The growth speed during runs for this simulated microtubule then followed the extrapolated curve as a function of force. We used a simple Monte Carlo-based method (again) to simulate traces of microtubule tip position versus time. All model parameters were derived directly from the experimental force-clamp data, with no adjustable ('free') parameters. As in our dual-trap experiments, simulated microtubule tip pairs were coupled with either soft or stiff elastic springs to a constant shared load, $F_{TOT}$ = 8 pN. Simulations of individual microtubules growing under constant force closely resembled force-clamp recordings of real microtubules *in vitro*. (Compare **Figure 6—figure supplement 1** to **Figure 1a**.)

The pausing model of microtubule growth provided significantly better fits to our dual-trap data than the non-pausing model. Like real microtubules in the dual-trap experiments, the simulated pairs of tips tended to drift apart under soft-coupled conditions, while under stiff-coupled conditions they stayed closer together (**Figure 6a**). After 300 s, the average tip separation for simulated soft-coupled pairs grew to 0.79±0.01 µm (mean ± SEM from $N$=10,000 simulations), while simulated stiff-coupled pairs had a lower average tip separation of 0.43±0.005 µm (mean ± SEM from $N$=10,000 simulations; **Figure 6b**), similar to the corresponding separations measured in dual-trap experiments (0.8±0.2 µm and 0.25±0.06 µm, respectively). Simulated pairs of microtubule tips under stiff-coupled conditions were also less likely to exceed a threshold separation of 0.8 µm (**Figure 6c**, dashed curves), similar to the survival analyses of microtubule pairs measured with the dual-trap assay (**Figure 6c**, solid

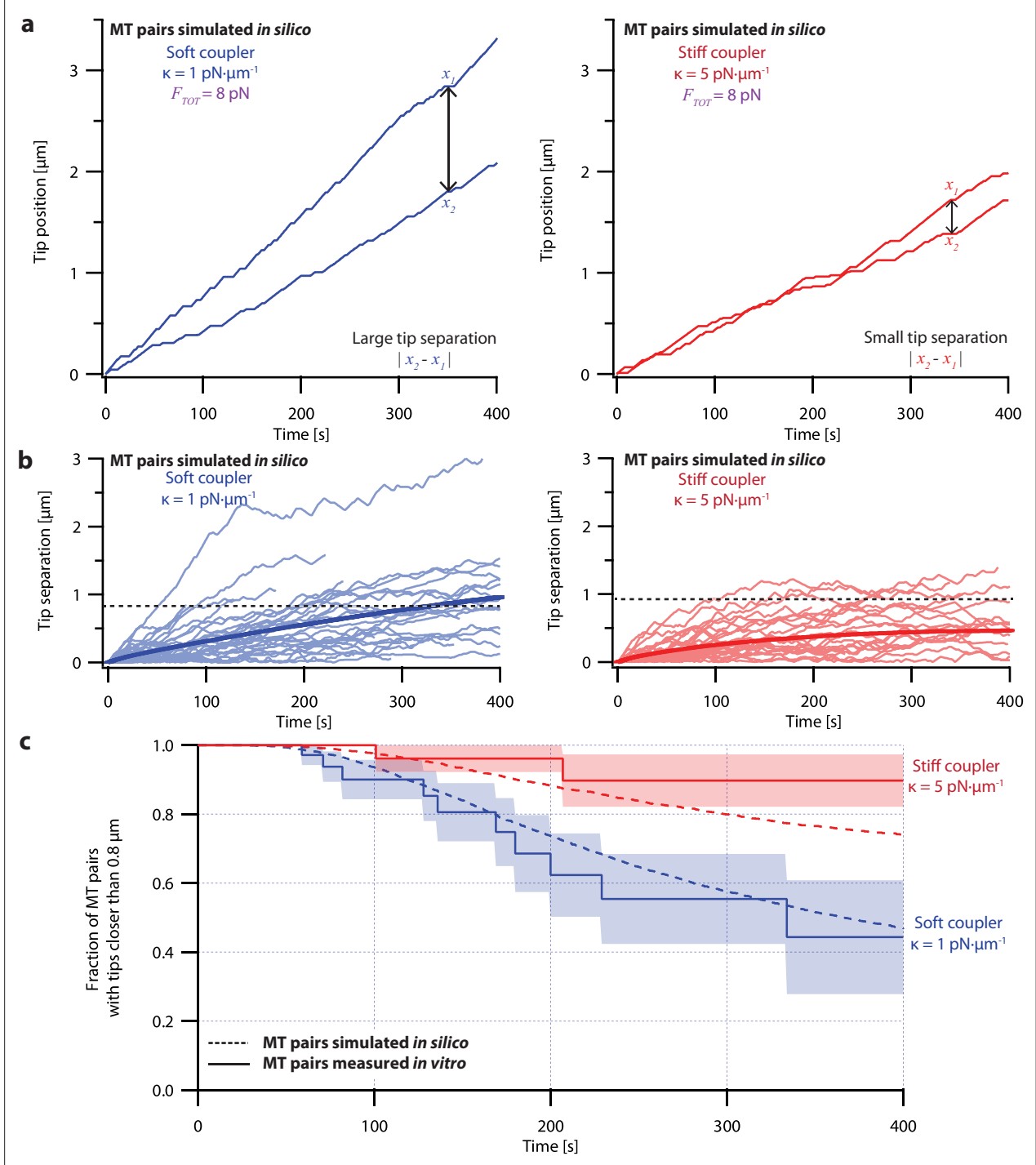

**Figure 6.** Simulations recapitulate the coordinated growth of mechanically coupled microtubule pairs. (**a**) Example simulations of microtubule (MT) plus-end positions over time using the pausing model. Simulated microtubule pairs were coupled via soft or stiff spring-couplers and shared a total load $F_{TOT}$ = 8 pN to match dual-trap assay conditions. (**b**) Tip separations between microtubule plus-end pairs that were simulated using the pausing model and coupled with either soft (left) or stiff (right) spring-couplers. Light colors show tip separations for $N$=40 individual simulated microtubule pairs, while dark curves show mean tip separation for $N$=10,000 simulated pairs. (**c**) Comparison between the fraction of *in silico* (simulated, dashed curves) and *in vitro* (real, solid curves) microtubule pairs whose plus-ends remained within 0.8 μm of each other over time. Shaded regions show SEMs from $N$=50 and $N$=43 *in vitro* recordings with soft and stiff couplers, respectively. Soft- and stiff-coupled microtubule pairs were each simulated $N$=10,000 times over 400 s of growth in both (**b**) and (**c**). Simulation parameters for (**a**), (**b**), and (**c**) are listed in *Table 1*.

The online version of this article includes the following figure supplement(s) for figure 6:

*Figure 6 continued on next page*

*Figure 6 continued*

**Figure supplement 1.** Simulations of microtubule growth under constant force using the pausing model resemble real microtubule recordings.

**Figure supplement 2.** Simulations of microtubule pairs using the pausing model maintain tip separations similar to real microtubule pairs.

**Figure supplement 3.** Simulations using an ergodic version of the pausing model, without intrinsic heterogeneity, fail to recapitulate the coordinated growth of mechanically coupled microtubules pairs.

**Figure supplement 4.** Probability analysis shows pausing model fits dual-trap recordings better than the non-pausing model.

curves). (Survival analyses of simulated microtubule pairs with threshold separations of 0.4, 0.6, and 1.2 µm also resembled the survival analyses of microtubule pairs measured with the dual-trap assay. See *Figure 6—figure supplement 2*). In addition, the pausing model scored better than the non-pausing model when we used probability to compare simulated to *in vitro* measurements (*Figure 6—figure supplement 4*). (To quantify the variability due to stochastic pausing alone, we also compared the pausing model to a version in which we assumed that microtubules were intrinsically identical ('ergodic'). The low variability predicted by this model (*Figure 6—figure supplement 3*), particularly under the soft-coupled condition, indicates that stochastic pausing alone is insufficient to generate the large and persistent differences in growth speed exhibited by real microtubules). Overall, the strong agreement between the pausing model and the dual-trap recordings suggests that mechanical coupling coordinates microtubule growth in at least two distinct ways: first, by counteracting persistent (intrinsic) differences in microtubule growth speed, and second, by selectively suppressing stochastic pausing in the lagging tip relative to the leading tip.

## Discussion

Our work demonstrates the effectiveness of mechanical coupling for coordinating the growth of microtubules and helps explain how numerous filaments within a k-fiber bundle can be tightly regulated to grow in near-perfect unison. Using our dual-trap assay, we found that pairs of microtubule tips growing *in vitro* are kept in close proximity when sharing a single load through a coupling material of sufficient stiffness. Tip pairs in our experiments were kept within a fraction of a micrometer by couplers of only $\kappa$=5 pN·µm⁻¹, a modest stiffness within the ranges reported for elements of the kinetochore (*Schwietert et al., 2022*; *Volkov et al., 2018*), pericentromeric chromatin (*Chacón et al., 2014*), and mitotic chromosomes (*Nicklas, 1983*; *Poirier et al., 2002*). Because the kinetochore and its underlying centromeric chromatin both appear to deform together to maintain close contacts with k-fiber plus-ends *in vivo* (*O'Toole et al., 2020*), the plus-ends presumably exert forces on each other through a composite material composed of both kinetochore and chromatin. We propose that this mechanical coupling is a major coordinating influence on plus-end growth dynamics within k-fiber bundles.

Our dual-trap approach provides information that prior single-trap experiments, and theoretical work based solely on these prior measurements, cannot provide. Microtubules in real k-fiber bundles presumably adjust their growth speeds dynamically in response to the forces they exert on each other. The dual-trap assay uniquely allows this kind of mutual dynamic influence to be examined. Because each microtubule in the pair grows simultaneously in the same preparation of buffer, any differences in growth speed represent true biological variability, without the potentially confounding influences of prep-to-prep or sample-to-sample variability. While previous experiments with individual microtubules implied that a shared load should tend to coordinate growing microtubules, the dual-trap experiments are a direct, model-independent way to examine the extent to which load sharing can suppress the intrinsic variability of microtubules growing under identical conditions. With the dual-trap assay, we can test models and assumptions about microtubule tension-dependence and variability derived from recordings of individual microtubules, as we did with the pausing and non-pausing models presented here.

Since most known eukaryotes have k-fibers of more than two microtubules, an important area of future work will be to study how effectively mechanical coupling can coordinate k-fibers with more microtubules. While extending our *in vitro* laser trap methods to more than two microtubules is impractical, our model can be easily extended to explore mechanical coordination within k-fibers of three or more microtubules. Simulation parameters, such as microtubule growth speeds, can also be changed to explore how well mechanical coupling coordinates microtubules in different cell types or

cell cycle stages with varying microtubule dynamics (*Cassimeris et al., 1988*; *Gaudenz et al., 2010*; *Yamashita et al., 2015*; *Rusan et al., 2001*). Similarly, simulations can be used to study how the addition of microtubule polymerases or depolymerases with known effects on microtubule dynamics influences coordination.

A variety of microtubule-modifying proteins are implicated in regulating k-fiber dynamics (reviewed in *Ou and Scholey, 2022* and *Amin et al., 2019*). These biochemical factors must work in concert with mechanical effects, and we suggest that the two classes of mechanisms probably operate over different length- and timescales. Certain tip-tracking proteins capable of influencing microtubule dynamics are enriched at growing k-fiber ends specifically near kinetochores that are moving anti-poleward (*Amaro et al., 2010*; *Castrogiovanni et al., 2022*; *Tirnauer et al., 2002*). Various kinesins are known to regulate the size of the metaphase spindle (*Goshima et al., 2005*; *Neahring et al., 2021*), to influence k-fiber dynamics (*Maiato et al., 2003a*; *Rizk et al., 2009*; *Stumpff et al., 2012*; *Wordeman et al., 2007*), and to alter microtubule dynamics *in vitro* in a length-dependent manner (*Varga et al., 2006*). However, it is unclear whether the specific enrichment of such microtubule-modulators on longer or faster-growing tips could occur quickly enough and with enough selectivity to slow the growth of a single tip that has grown ahead of its neighbors by only a fraction of a micrometer, as required for tight coordination within a k-fiber. On the other hand, mechanical forces would be transmitted essentially instantaneously between neighboring tips through local deformations of the material through which the tips are coupled. Therefore, mechanical coupling could account for the rapid, sub-micron responses required to coordinate microtubule tips within a k-fiber, while microtubule-modifiers and spindle-wide phenomena, like polar ejection forces and poleward flux, are likely to help set overall k-fiber length (*Risteski et al., 2021*).

We found that tension accelerates microtubule growth by not only directly increasing growth speed but also decreasing time spent in pause. More specifically, tension reduced the rate at which growing tips entered a pause state. This finding implies that pause entry is associated with movement of tension-bearing elements at the tip toward the minus end, opposite the direction of applied force and therefore inhibited by the force. Tension also increased the rate at which paused tips resumed growth, which implies that pause exit is associated with movement toward the plus-end, in the same direction as the applied force and therefore assisted by the force. These tension-dependent dynamics may reflect mechanical deformations of the microtubule tip; perhaps pauses occur when newly added tubulin dimers at a growing tip adopt curved structures that disfavor lateral bonding (*McIntosh et al., 2018*). Tension could help straighten these curved structures, thereby accelerating lateral bond formation and favoring growth (*Cleary and Hancock, 2021*).

We derived a model of microtubule growth with no free parameters, based on constant-force recordings of individual microtubules (from *Akiyoshi et al., 2010*), and found that a combination of stochastic, force-dependent pausing and intrinsic differences in microtubule growth speed explained both the variability and the coordination measured in our dual laser trap assay. The necessity of including microtubules with different intrinsic growth speeds for our model's success implies structural heterogeneity in the growing tips that must persist for many minutes, even as thousands of tubulin subunits are being added. In principle, these persistent differences might arise from different or mismatched protofilament numbers (*Chrétien et al., 1992*; *Rai et al., 2021*) or the accumulation of other lattice defects (*Wieczorek et al., 2015*) that result in varying degrees of tip taper (*Coombes et al., 2013*; or tip 'raggedness' *Brouhard and Rice, 2018*) which are more or less conducive to microtubule growth.

For technical reasons, we focused exclusively here on microtubule tips in a state of net growth. However, coordination of k-fiber microtubules *in vivo* presumably also requires tight regulation of tip shortening speeds and rates of switching between growth and shortening (i.e. 'catastrophe' and 'rescue' rates; *Banigan et al., 2015*; *Schwietert and Kierfeld, 2020*). These will be more difficult to study with our dual-trap assay because switch events are stochastic and episodes of shortening are observed less frequently than episodes of growth under our assay conditions. Nevertheless, the dual-trap approach can be used in the future (potentially with a cutting laser to sever the microtubule tips to induce shortening *Driver et al., 2017*; *Murray et al., 2022*) to study how mechanical coupling affects microtubule pairs in which one or both filaments are shortening, as well as switch rates between shortening and growth. Our dual-trap approach can also be adapted to examine mechanical coupling through materials with complex mechanical properties, such as viscoelasticity and strain-stiffening

(*Cojoc et al., 2016*; *Schwietert et al., 2022*). In principle, the effect of mechanical coupling on pairs of processive motor proteins (*Holzbaur and Goldman, 2010*) or on pairs of dynamic actin filaments linked to a shared load (*Garner and Theriot, 2022*) can also be studied with the dual-trap approach. Overall, our work demonstrates that mechanical coupling is a feasible mechanism that could coordinate k-fiber microtubules and maintain their synchronous growth in the mitotic spindle.

## Methods

### Spring-coupler model

For both *in vitro* and *in silico* experiments, the force on each microtubule plus-end was dictated by a model in which plus-ends were coupled to a shared 'kinetochore' via simple Hookean springs of identical stiffnesses (*Figure 2b*). Because k-fibers grow relatively slowly in cells (*Khodjakov et al., 1996*; *Skibbens et al., 1993*; *Stumpff et al., 2012*), we neglect viscous drag such that the sum of the forces on individual k-fiber tips equals $F_{TOT}$. Microtubule pairs were considered to grow along a shared 1D axis. Microtubules were assumed to be incompressible because microtubules have high axial stiffness (*Howard, 2001*; *Kis et al., 2002*; *Pampaloni et al., 2006*) and are cross-linked in bundles (*O'Toole et al., 2020*), thus forming the structural support required to maintain mitotic spindle integrity. Microtubules were considered stationary except at the plus-end, so that only growth or shortening at the plus-end affected plus-end position.

### Kinetochore purification

Native kinetochore particles were purified from asynchronously grown *S. cerevisiae* SBY8253 cells (Genotype: MATa DSN1-6His-3Flag:URA3) grown in YPD medium (2% glucose) by modifying previous protocols (*Akiyoshi et al., 2010*; *Miller et al., 2016*). Protein lysates were prepared using a Freezer Mill (SPEX SamplePrep) submerged in liquid nitrogen. Lysed cells were resuspended in buffer H (25 mM Hepes, pH 8.0, 2 mM $MgCl_2$, 0.1 mM EDTA, 0.5 mM EGTA, 0.1% NP-40, 15% glycerol, and 150 mM KCl) containing phosphatase inhibitors (0.1 mM Na-orthovanadate, 0.2 µM microcystin, 2 mM β-glycerophosphate, 1 mM Na pyrophosphate, and 5 mM NaF) and protease inhibitors (20 µg/ml leupeptin, 20 µg/ml pepstatin A, 20 µg/ml chymostatin, and 200 µM PMSF). Lysates were ultracentrifuged at 98,500 *g* for 90 min at 4 °C. Protein G Dynabeads (Invitrogen) were conjugated with an α-FLAG antibody (Sigma-Aldrich), and immunoprecipitation of Dsn1-6His-3Flag was performed at 4 °C for 3 hr. Beads were washed once with lysis buffer containing 2 mM DTT and protease inhibitors, three times with lysis buffer with protease inhibitors, and once in lysis buffer without inhibitors, and kinetochore particles were eluted by gentle agitation of beads in elution buffer (buffer H+0.5 mg/ml 3FLAG Peptide [Sigma-Aldrich]) for 30 min at room temperature.

### Bead functionalization and slide preparation for laser trap experiments

To functionalize laser trapping beads with native kinetochore particles, streptavidin-coated polystyrene beads (0.44 µm in diameter, Spherotech) were incubated with biotinylated anti-5His antibodies (Qiagen) and stored for up to 6 months with continuous rotation at 4 °C in BRB80 (80 mM Pipes, 1 mM $MgCl_2$, and 1 mM EGTA, pH 6.9) supplemented with 8 mg·mL$^{-1}$ bovine serum albumin and 1 mM DTT. Before each experiment, functionalized beads were sonicated for two minutes and decorated with kinetochore particles by incubating 6 pM anti-5His beads for 1–2 hr at 4 °C with purified kinetochore material (in BRB80 with 1 mg·mL$^{-1}$ K-casein), corresponding to concentrations of Dsn1-His-Flag between 1.5 and 4 nM, well within the single-kinetochore-particle condition described in *Akiyoshi et al., 2010*.

Each flow chamber (~5–10 µL in volume) was constructed by placing two parallel strips of double-stick tape between a glass slide and a KOH- or plasma-cleaned coverslip. Immediately before every experiment, two flow chambers were each functionalized as follows. First, flow chambers were incubated with 10 µL of 10 mg·mL$^{-1}$ biotinylated bovine serum albumin (Vector Laboratories) for 15 min at room temperature. Chambers were then washed with 100 µL BRB80. Next, 20 µL of 333 mg·mL$^{-1}$ avidin DN (Vector Laboratories) in BRB80 was added to the chamber and incubated for 3 min before washing again with 100 µL BRB80. Chambers were then incubated for 5 min with GMPCPP-stabilized biotinylated microtubule seeds warmed for 5–10 min in BRB80. Next, chambers were washed with 100 µL of growth buffer (1 mM GTP, 1 mg·mL$^{-1}$ K-casein, and 0.08 mg·mL$^{-1}$ biotinylated bovine serum

albumin in BRB80). Finally, chambers were incubated with kinetochore-decorated beads diluted approximately eightfold into growth buffer augmented with 1.5 mg·mL$^{-1}$ purified bovine brain tubulin and an oxygen scavenging system (1 mM DTT, 500 µg·mL$^{-1}$ glucose oxidase, 60 µg·mL$^{-1}$ catalase, and 25 mM glucose). Flow chamber exits and entrances were then sealed with nail polish. Chambers were used for experiments at 22°C for up to 2 hr. The tubulin used in all the dual-trap experiments came from a single preparation, so all $N$=93 recordings represent technical replicates.

## Performing the dual-trap assay

Laser trap experiments were performed using two custom-built instruments based on commercial inverted microscopes (Nikon; TE2000U) modified to each incorporate a trapping laser (Spectra-Physics; J20I-BL-106C-NSI), a computer-controlled piezo specimen stage (Physik Instrumente; P-517.3CL), and a position-sensitive photodetector (Pacific Silicon Sensor; DL100-7-PCBA2). Experiments were performed at 22 ° C with trap stiffnesses of 0.35–0.91 pN·nm$^{-1}$. See (*Franck et al., 2010*) for details regarding instrument setup and calibration.

The dual-trap assay was modified from previous force-clamp protocols (*Akiyoshi et al., 2010*; *Franck et al., 2010*; *Miller et al., 2016*) and maintained constant total force on two bead-laser trap systems, rather than one (*Figure 2a*). Both lasers were controlled by a single computer, which adjusted the force on each bead to maintain force equilibrium on the modeled coupler given the measured bead positions (*Figure 2b*). The bead positions served as accurate proxies for the microtubule tip positions because kinetochore-decorated beads are known to track with microtubule plus-ends under tensions between 0.5 and 18 pN (*Akiyoshi et al., 2010*; *Asbury et al., 2006*; *Franck et al., 2010*; *Miller et al., 2016*; *Powers et al., 2009*; *Tien et al., 2010*). In addition, the yeast kinetochores used to link the beads to the microtubule tips are very small, only ~130 nm in length (*Gonen et al., 2012*), compared to the relevant micrometer-scale tip displacements. The small size of the yeast kinetochore, as well as the slow change in force on each tip (*Figure 3—figure supplement 1*), ensures that the kinetochore's (visco)elastic properties did not significantly affect experimental results, because any force-dependent changes in its length had a negligible effect compared to tip growth.

To calibrate the laser trap's zero-force position, we recorded the position of a free bead using each laser-trap at the start of each 2 hr experiment and as necessary between recordings. Feedback to precisely exert the amount of force dictated by the relative bead positions and the spring-coupler model was implemented with custom LabView software (National Instruments). Bead-trap separation and stage position were sampled at 40 kHz, while stage position to maintain the desired bead-trap separation (and thus force) was updated at 50 Hz. Bead and stage position data were decimated to 200 Hz before storing to disk, a temporal frequency which is sufficient to capture pauses and runs of many seconds.

LabView software to perform force-clamp experiments is available at https://github.com/casbury69/laser-trap-control-and-data-acquisition, (*Asbury, 2022*) and the LabView software to perform dual-trap experiments is available at https://github.com/casbury69/dual-trap-control-and-data-acquisition, (*Asbury and Leeds, 2023*).

## Analyzing dual-trap recordings

We analyzed recordings from the dual-trap assay using custom software written in Igor Pro 9 (Wavemetrics). First, by hand, we marked the beginning and end of recorded segments in which both microtubules were growing and separated these episodes into individual recordings. We then calculated the tip separation between coupled microtubules for each excised recording by taking the absolute value of the difference in position at 1 s intervals from 0 to 400 s. For each recording, at 0.005 s intervals, the force on each microtubule was calculated by multiplying the trap stiffness by the distance between the current and unloaded bead position, while total force on the microtubule pair was computed by adding the force on both microtubules together (*Figure 3—figure supplement 1*).

To calculate the speed of tip separation, we fit the first 100 s to a line with the Levenberg-Marquardt method. Uncertainties in tip separation speed were estimated from residuals as described in *Press et al., 1992*.

SEMs for microtubule pair tip separation measured using the dual-trap assay (*Figure 3b*) were calculated at 1 s intervals after the recording began using the following equation:

$$SEM_{tip\ sep}\left(t\right) = \frac{\sigma_t}{\sqrt{N_t}}$$

where $\sigma_t$ is the standard deviation of the tip separations at t seconds and $N_t$ is the number of microtubule pairs which were still growing and tip-attached to a bead at t seconds.

Download files at https://doi.org/10.5061/dryad.xksn02vnb to view position- and force-versus-time curves for each of the dual-trap recordings analyzed in this work. See also 'Dual-trap anlaysis.txt' at https://doi.org/10.5281/zenodo.8433109 for the software used to analyze dual-trap recordings.

### Creating Kaplan-Meier survival curves

Kaplan-Meier survival curves for both simulated and *in vitro* microtubule pairs (*Figure 3c*, *Figure 4c*, *Figure 6c*, *Figure 6—figure supplement 2*, and *Figure 6—figure supplement 3c*) were calculated at 1 s intervals after the recording began using the following equation:

$$S\left(t\right) = \prod_{j=0}^{t} \frac{n_j - d_j}{n_j}$$

where $S\left(t\right)$ is the Kaplan-Meier survival probability at $t$ seconds, $t$ is the number of Kaplan-Meier estimates made between time 0 and $t$ seconds, $n_j$ is the number of microtubule pairs still growing, tip-attached to a bead, and within the threshold tip separation at $j$ seconds, and $d_j$ is the number of microtubule pairs that exceeded the threshold tip separation (0.4, 0.6, 0.8, or 1.2 µm) between time $j - 1$ and $j$ seconds.

SEMs for Kaplan-Meier survival plots estimated using the dual-trap assay (*Figure 3c* and *Figure 6—figure supplement 2*) were calculated at 1 s intervals after the recording began using the following equation:

$$SEM_{survival}\left(t\right) = S\left(t\right) * \sqrt{\sum_{j=1}^{t} \frac{d_j}{n_j\left(n_j - d_j\right)}}$$

where $S\left(t\right)$ is the Kaplan-Meier survival probability at $t$ seconds, $t$ is the number of Kaplan-Meier estimates made between time 0 and $t$ seconds, $d_j$ is the number of microtubule pairs that exceeded the threshold tip separation (0.4, 0.6, 0.8, or 1.2 µm) between time $j - 1$ and $j$ seconds, and $n_j$ is the number of microtubule pairs still growing, tip-attached to a bead, and within the threshold tip separation at $j$ seconds.

### Segmenting constant force traces into periods of run and pause

To estimate pause entrance and exit rates as a function of force, we segmented $N$=356 recordings of growing microtubules under constant force (from *Akiyoshi et al., 2010*) into pauses and fast growth ('runs,' *Figure 5—figure supplement 1a and b*). First, we estimated the instantaneous growth speed over the entirety of each recording to be the slope of the best fit line for a sliding 2 s window (*Figure 5—figure supplement 1c*). Next, we defined a unique threshold between pause and run for each recording by assembling a histogram of growth speeds throughout the recording and fitting that histogram with a sum of between two and six Gaussian peaks. We centered one Gaussian at or near 0 µm·min⁻¹ and defined it as the 'pause' peak. Fitting the growth speed histograms to Gaussian peaks was performed using the 'Multipeak fitting' package in Igor Pro 9. The intersection of the pause peak and the Gaussian with the lowest non-zero mean growth speed was identified for each individual recording and defined as the threshold to distinguish pauses and runs for that recording.

To reduce the impact of high-frequency noise, we excluded pauses and runs that lasted less than 1 s. If the instantaneous growth speed fell below that recording's predefined threshold for at least 1 s, the microtubule was considered to have paused. If the instantaneous growth speed fell below the threshold for less than 1 s, the short interval was combined with the run intervals before and after to make a single run. Likewise, if the instantaneous growth speed exceeded the threshold for at least 1 s, the microtubule was considered to be in run. If the instantaneous growth speed exceeded the threshold for less than 1 s, the short interval was combined with the pause intervals before and after to make a single pause. If the defined threshold was never crossed, the recording was considered a single run interval.

The overall trends of tension suppressing pause entrance and promoting pause exit were maintained irrespective of our choices of minimum event duration and smoothing window size. While the 1 s minimum event duration decreased the overall frequency of pause entrances and exits, it did not significantly affect the measured force sensitivities of pause entrance ($F_{EN}$) and exit ($F_{EX}$). Re-analyzing the constant force recordings with only a 0.01 s minimum event duration, we measured force sensitivities of $F_{EN}$ = –5.1 pN and $F_{EX}$ = 14.3 pN, compared to $F_{EN}$ = –4.6 pN and $F_{EX}$ = 14.1 pN with the 1 s minimum event duration. The size of the smoothing window had a significant effect on measured pause exit force sensitivity, but not pause entrance force sensitivity—re-analyzing the constant force recordings with a 5 s smoothing window, we measured force sensitivities of $F_{EN}$ = –4.4 pN and $F_{EX}$ = 24.8 pN. Ultimately, we chose the 2 s smoothing window and 1 s minimum duration threshold because these choices resulted in recording segmentation that agreed best with how we would differentiate pauses and runs by eye.

Download files at https://doi.org/10.5061/dryad.6djh9w16v to view position- and velocity-versus time curves for each force-clamp recording analyzed in this work, as well as how pauses were marked for each recording. This file also includes additional shortening events which were not studied here. See also 'Force-clamp viewer.txt' at https://doi.org/10.5281/zenodo.8433062 for the software used to distinguish pauses from runs.

## Quantifying growth speed during runs for individual recordings

For each pause-based model that we considered, microtubule growth speed during runs was assumed to be time-invariant under constant force. Thus, we made the same assumption when we quantified the growth speed during runs of individual microtubules *in vitro*, and calculated a unique growth speed during runs for each recording (dots in *Figure 5b*). The growth speed during each run interval was calculated by averaging together the instantaneous growth speeds comprising that interval, measured with the 2 s sliding window described above. We calculated the overall growth speed during runs for a single microtubule by averaging together the growth speed of all contained run intervals, weighted by the duration of that interval, such that:

$$v_{run} = \frac{\sum v_i * t_i}{T}$$

where $v_{run}$ is the overall growth speed during runs for a particular recording, $v_i$ is the growth speed for the $i^{th}$ run interval, $t_i$ is the duration of the $i^{th}$ run interval, and $T$ is the total time spent in the run state.

## Quantifying the force dependence of pause entrance rates, exit rates and growth speed

To calculate average pause exit rates, entrance rates, and growth speeds as a function of force we binned each recording by the constant tensile force to which the microtubule was subjected (*Figure 5*).

**Table 2.** Number of recordings, bounds, and pause quantification from force-clamp dataset, binned by force.

Table describing constant force recordings in *Akiyoshi et al., 2010* and the pause quantification performed in the current work. *N* indicates the number of recordings in a particular bin. For consistency, these force bins are identical to those used in *Akiyoshi et al., 2010*.

| Bin number | N | Lower bound [pN] | Upper bound [pN] | Pause entrances | Time in pause [s] | Pause exits | Time in run [s] |
|---|---|---|---|---|---|---|---|
| 1 | 41 | 0.69 | 1.94 | 1,834 | 10,957 | 1,830 | 7,364 |
| 2 | 85 | 1.94 | 2.97 | 3,502 | 18,047 | 3,501 | 18,952 |
| 3 | 77 | 2.97 | 3.999 | 2,376 | 12,842 | 2,374 | 18,159 |
| 4 | 60 | 3.999 | 5.98 | 1,937 | 9,661 | 1,932 | 16,993 |
| 5 | 60 | 5.98 | 7.99 | 1,439 | 5,930 | 1,438 | 18,465 |
| 6 | 25 | 7.99 | 10.3 | 387 | 1,298 | 378 | 6,559 |
| 7 | 9 | 10.3 | 18 | 23 | 93 | 23 | 712 |

For consistency, we used the same seven force bins used by *Akiyoshi et al., 2010* (see *Table 2* for bin boundaries). For both the pause entrance and exit rate, we excluded the highest force bin from our analysis because of low statistics. For each force bin, we calculated the mean pause entrance and exit rate by counting the number of times microtubules in that force bin entered and exited pause and dividing by the total time microtubules in that bin spent in run and pause, respectively (*Figure 5—figure supplement 1a*). Error bars in the y-direction represent counting uncertainties, calculated with the equation:

$$SEM_{rate}\left(i\right) \ = \ \frac{k_i}{\sqrt{N_i}}$$

where $k_i$ is the rate of pause entrance or exit for the $i^{th}$ force bin, and $N_i$ is the number of pause entrances or exits in the recordings in the $i^{th}$ bin ($N$=387–3,502 pause entrances and $N$=378–3,501 pause exits per bin, see *Table 2* for the number of pause entrances and exits by bin). Error bars in the x-direction represent standard deviations of forces for the recordings in each bin.

We found the mean growth speed during runs for each force bin by averaging together the growth speed during runs for all the recordings in that bin. Error bars in the y-direction represent the SEMs of growth speeds, calculated with the equation:

$$SEM_{run\ speed}\left(i\right) \ = \ \frac{\sigma_i}{\sqrt{N_i}}$$

where $\sigma_i$ is the standard deviation of the growth speeds during runs, averaged over each recording, for the $i^{th}$ force bin, and $N_i$ is the number of recordings in the $i^{th}$ bin ($N$=9–85 recordings per bin, see *Table 2* for the number of recordings by bin). Once again, error bars in the x-direction represent the standard deviations of forces for the recordings in each bin. Fit lines for pause exit and entrance rates and growth speed were calculated with Levenberg-Marquardt least-squares method (*Press et al., 1992*).

## Simulating pairs of growing microtubules

Simulated recordings using each model followed the same general procedure. For simulations in which the growth or run speed was different between microtubules (both non-ergodic models), the 'unloaded' growth or run speed was drawn from a distribution of measured growth or run speeds as described in the results section. For the non-ergodic pausing model, intrinsic unloaded run speeds ($v_{intrinsic}$) of more than 30 nm/s were considered outliers and excluded from the distribution. For microtubules simulated with the ergodic pausing model, both microtubules had the same growth speed during runs, based on the equation and parameters for $v_{run}$ in *Table 1*. Each simulation began with both microtubules growing (in run, for pause-based models) from the same position under the same tensile force. As microtubules stochastically grew or paused, the tensile force on each microtubule evolved according to the spring-coupler model, just like they did in the dual-trap assay. As the force changed, depending on the model type, the growth or run speed changed according to exponential relationships with force estimated from *Akiyoshi et al., 2010*; *Table 1*.

A modified Gillespie algorithm was used to implement stochastic microtubule events: pause entrance and exit, detachment from the kinetochore, switches into shortening (catastrophe), and other event interruptions. Each of these event rates were assumed independent Poisson processes and were derived directly from the data summarized in *Akiyoshi et al., 2010*. Each microtubule was assumed to have identical rates for all stochastic events, given the same external force. Detachment and catastrophe rates were calculated using the exponential relationships as a function of force estimated by *Akiyoshi et al., 2010*. The rate of interruption by other events was also estimated from these force-clamp recordings but was force-independent. For simulations that included pausing, rates of pause entrance and exit were calculated from the fit equations in *Table 1* (fit lines in *Figure 5*).

Because detachment, catastrophe, and pause rates are force-dependent, they can only be approximated as Poisson (time-invariant) rates for short time intervals over which the force on the tip is relatively constant. Consequently, we imposed a maximum timestep of no greater than 1 s for each simulation. If the timestep chosen by our Gillespie algorithm exceeded the maximum timestep, no event occurred and the microtubule continued growing during that 1 s interval. All simulations were performed using custom software in Igor Pro 9 (Wavemetrics).

Download 'Dual-trap simulations.pxp' at https://github.com/casbury69/simulations-of-coupled-microtubules, (*Leeds and Asbury, 2023*) to run dual-trap simulations as described in the results and see 'Dual-trap simulator.txt' at https://github.com/casbury69/simulations-of-coupled-microtubules for the code used to run these simulations.

## Comparing pausing and non-pausing models with simple probability

To evaluate whether the pausing or non-pausing model was more likely given our dual-trap measurements, we compared histograms of tip separations at 100 and 200 s after the start of the recording or simulation (*Figure 6—figure supplement 4*). We used 0.1 μm bins for both simulated and dual-trap data and normalized all histograms to have an area of 1.

We calculated the likelihood that simulations performed using the pausing and non-pausing models would produce each *in vitro* tip separation:

$$L_{measurement} = w_{bin} * P_{normalized}$$

where $w_{bin}$ is the bin width of the tip separation histogram (0.1 μm) and $P_{normalized}$ is the normalized probability of the given model at the measured tip separation (y-values of the dashed lines in *Figure 6—figure supplement 4*).

To compare the overall likelihood that the pausing and non-pausing simulations would produce all the tip separations measured *in vitro*, we multiplied together the likelihoods of each measurement for a given model at a given coupler stiffness and at a given time point. We calculated the log likelihood by taking the base 10 logarithm of each probability (reported in the right-hand column of each table in *Figure 6—figure supplement 4*).

## Statistical analysis

All statistical analysis was performed using custom software written in Igor Pro 9 (Wavemetrics).

We calculated p-values to determine whether survival curves of stiff-coupled and soft-coupled *in vitro* microtubule pairs were statistically different for each threshold tip separation using a log-rank test. For each survival curve, at 1 s intervals after the recording began, we calculated an expected value (the expected number of microtubule pairs to exceed the threshold tip separation between time $t-1$ and t seconds) using the following equation:

$$E\left(t\right) = \frac{n_t}{n_{tot}}\left(d_{tot}\right)$$

where $n_t$ is the number of microtubule pairs comprising one survival curve that are still growing, tip-attached to a bead, and within the threshold tip separation at $t$ seconds, $n_{tot}$ is the total number of microtubule pairs comprising both survival curves being compared that are still growing, tip-attached to a bead, and within the threshold tip separation at $t$ seconds, and $d_{tot}$ is the total number of microtubule pairs that exceeded the threshold tip separation between time $t-1$ and $t$ seconds for both survival curves being compared. Expected values for each pair of survival curves were used to calculate a chi-square value for each tip separation threshold using the following equation:

$$\chi^2 = \sum_{j=1}^{2} \frac{\left(\sum_{t=1}^{T} d_j\left(t\right) - \sum_{t=1}^{T} E_j\left(t\right)\right)^2}{\sum_{t=1}^{T} E_j\left(t\right)}$$

where $T$ is the total number of timepoints (400, in all cases), $d_j\left(t\right)$ is the observed number of microtubule pairs that exceeded the threshold tip separation for Kaplan-Meier survival curve $j$ between time $t-1$ and $t$ seconds, and $E_j\left(t\right)$ is the expected number of microtubule pairs to exceed the threshold tip separation for Kaplan-Meier survival curve $j$ between time $t-1$ and t seconds. A chi-square table was then used to estimate the *p*-value for each tip separation threshold.

To determine whether tip separation and tip separation speed between soft and stiff couplers were statistically significant, we calculated *p*-values with a Wilcoxon-Mann-Whitney two sample rank test using built-in Igor Pro software (*Cheung and Klotz, 1997*; *Klotz, 2006*; *Streitberg and Rohmel, 1986*; *Zar, 1999*). The Wilcoxon-Mann-Whitney two sample rank test does not assume that the two populations being compared are Gaussian (nonparametric), which makes it more appropriate than

parametric tests for analyzing dual-trap data, for which we do not necessarily expect normal distributions of tip separations.

## Acknowledgements

We are grateful for feedback and critical reading of the manuscript from members of the Biggins, Asbury, and Trisha Davis laboratories.

This work was supported by a Packard Fellowship 2006–30521 (to CL Asbury), National Institutes of Health grants R01GM079373, P01GM105537, and R35GM134842 (to CL Asbury) and R01GM064386 (to S Biggins), and by the Genomics and Scientific Imaging and the Proteomics and Metabolomics Shared Resources of the Fred Hutchinson/University of Washington Cancer Consortium (P30 CA015704). S Biggins is an investigator of the Howard Hughes Medical Institute. Research reported in this publication was also supported by the National Institute of General Medical Sciences of the National Institutes of Health under Award Number T32GM008268 (to BK Leeds).

## Additional information

### Funding

| Funder | Grant reference number | Author |
|---|---|---|
| David and Lucile Packard Foundation | Packard Fellowship 2006-30521 | Charles L Asbury |
| National Institutes of Health | R01GM079373 | Charles L Asbury |
| National Institutes of Health | P01GM105537 (Asbury,CoI; Mark Winey,PI) | Charles L Asbury |
| National Institutes of Health | R35GM134842 | Charles L Asbury |
| National Institutes of Health | R01GM064386 | Sue Biggins |
| Fred Hutchinson/University of Washington Cancer Consortium | Genomics and Scientific Imaging and the Proteomics and Metabolomics Shared Resources P30 CA015704 | Sue Biggins |
| National Institutes of Health | Graduate student training position T32GM008268 | Bonnibelle K Leeds |
| Howard Hughes Medical Institute | Sue Biggins is an HHMI investigator | Sue Biggins |

The funders had no role in study design, data collection and interpretation, or the decision to submit the work for publication.

### Author contributions

Bonnibelle K Leeds, Conceptualization, Data curation, Software, Formal analysis, Supervision, Funding acquisition, Validation, Investigation, Visualization, Writing – original draft, Writing – review and editing; Katelyn F Kostello, Validation, Investigation, Visualization, Writing – review and editing; Yuna Y Liu, Data curation, Software, Formal analysis, Validation, Visualization; Christian R Nelson, Resources, Validation; Sue Biggins, Resources, Supervision, Funding acquisition, Writing – review and editing; Charles L Asbury, Conceptualization, Resources, Software, Supervision, Funding acquisition, Validation, Investigation, Visualization, Writing – original draft, Project administration, Writing – review and editing

### Author ORCIDs

Bonnibelle K Leeds http://orcid.org/0009-0009-8932-707X
Christian R Nelson http://orcid.org/0000-0001-8717-0229

Sue Biggins [ORCID] http://orcid.org/0000-0002-4499-6319
Charles L Asbury [ORCID] http://orcid.org/0000-0002-0143-5394

Reviewer #1 (Public Review): https://doi.org/10.7554/eLife.89467.3.sa1
Reviewer #2 (Public Review): https://doi.org/10.7554/eLife.89467.3.sa2
Author Response https://doi.org/10.7554/eLife.89467.3.sa3

## Additional files

### Supplementary files
• MDAR checklist

### Data availability
All the position- and force-versus-time curves for each of the dual-trap recordings analyzed in this work have been deposited onto the DRYAD repository at https://doi.org/10.5061/dryad.xksn02vnb. All the position- and velocity-versus time curves for each of the single-trap force-clamp recordings analyzed in this work, as well as the markup of pauses within these recordings, have been deposited onto the DRYAD repository at https://doi.org/10.5061/dryad.6djh9w16v.

The following datasets were generated:

| Author(s) | Year | Dataset title | Dataset URL | Database and Identifier |
|---|---|---|---|---|
| Leeds BK, Kostello KF, Liu YY, Biggins S, Asbury CL | 2023 | Paired microtubules growing with a shared load | https://doi.org/10.5061/dryad.xksn02vnb | Dryad Digital Repository, 10.5061/dryad.xksn02vnb |
| Bungo A, Sarangapani KK, Powers AF, Nelson CR, Reichow SL, Arellano-Santoyo H, Gonen T, Ranish JA, Leeds B, Asbury CL, Biggins S | 2023 | Microtubules growing and shortening under constant force | https://doi.org/10.5061/dryad.6djh9w16v | Dryad Digital Repository, 10.5061/dryad.6djh9w16v |

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
