## [Editor Report · eLife assessment]

In this technically advanced and **important** piece of work, the authors study the coordination of microtubule growth in kinetochore fibers using force spectroscopy and numerical simulations. With **compelling** evidence the authors address the question of how microtubules, which naturally exhibit variable growth rates, can coordinate their behavior by mechanical coupling so as to function as a single unit in generating forces during chromosome segregation.

---

## [Referee Report · Reviewer #1 (Public Review)]

This remarkable and creative study from the Asbury lab examines the extent to which mechanical coupling can coordinate the growth of two microtubules attached to isolated kinetochores. The concept of mechanical coupling in kinetochores was proposed in the mid-1990s and makes sense intuitively (as shown in Fig. 1B). But intuitive concepts still need experimental validation, which this study at long last provides. The experiments described in this paper will serve as a foundation for the transition of an intuitive concept into a robust, quantitative, and validated model.

The introduction cites at least 5 papers that proposed mechanical coupling in kinetochores, as well as 5 theoretical studies on mechanical coupling within microtubule bundles, so it's clear that this manuscript will be of considerable interest to the field. The intro is very well written (as is the manuscript in general), but I recommend that the authors include a brief review of the variable size of k-fibers across species, to help the reader contextualize the problem. For example, budding yeast kinetochores are built around a single microtubule (Winey 1995), so mechanical coupling is not relevant for this species.

Indeed, the use of yeast kinetochores to study mechanical coupling is an odd fit, because these structures did not evolve to support such coupling. There is no doubt that yeast kinetochores are useful for demonstrating mechanical coupling and for measuring the stiffnesses necessary to achieve coupling, but I recommend that the authors include a caveat somewhere in the manuscript, perhaps in the place where they discuss their use of simple elastic coupling as compared to viscoelastic coupling or strain-stiffening. It's easy to imagine that kinetochores with large k-fibers might require complex coupling mechanisms, for example. And is mechanical coupling relevant for holocentric kinetochores like those found in *C. elegans*?

The paper shows considerable rigour in terms of experimental design, statistical analysis, and presentation of results. My only comment on this topic relates to the bandwidth of the dual-trap assay, which I recommend describing in the main text in addition to the methods. For example, the authors note that the stage position is updated at 50 Hz. The authors should clearly explain that this bandwidth is sufficiently fast relative to microtubule growth speeds.

After describing their measurements, the authors use Monte Carlo simulations to show that pauses are essential to a quantitative explanation of their coupling data. Apparently, there is a history of theoretical approaches to coupling, as the introduction cites 5 theoretical studies.

Overall, this paper is rigorous, creative, and thought-provoking. The unique experimental approach developed by the Asbury lab shows great promise, and I very much look forward to future iterations.

---

## [Referee Report · Reviewer #2 (Public Review)]

Leeds et al. employ elegant *in vitro* experiments and sophisticated numerical modeling to investigate the ability of mechanical coupling to coordinate the growth of individual microtubules within microtubule bundles, specifically k-fibers. While individual microtubules naturally polymerize at varying rates, their growth must be tightly regulated to function as a cohesive unit during chromosome segregation. Although this coordination could potentially be achieved biochemically through selective binding of polymerases and depolymerases, the authors demonstrate, using a novel dual laser trap assay, that mechanical coupling alone can also coordinate the growth of *in vitro* microtubule pairs.

By reanalyzing recordings of single microtubules growing under constant force (data from their own previous work), the authors investigate the stochastic kinetics of pausing and show that pausing is suppressed by tension. Using a constant shared load, the authors then show that filament growth is tightly coordinated when pairs of microtubules are mechanically coupled by a material with sufficient stiffness. In addition, the authors develop a theoretical model to describe both the natural variability and force dependence of growth, using no freely adjustable parameters. Simulations based on this model, which accounts for stochastic force-dependent pausing and intrinsic variability in microtubule growth rate, fit the dual-trap data well.

Overall, this study illuminates the potential of mechanical coupling in coordinating microtubule growth and offers a framework for modeling k-fibers under shared loads. The research exhibits meticulous technical rigor and is presented with exceptional clarity. It provides compelling evidence that a minimal, reconstituted biological system can exhibit complex behavior. As it currently stands, the paper is highly informative and valuable to the field.

---

## [Author Response]

The following is the authors’ response to the original reviews.

We are grateful to all the reviewers for their thoughtful comments and the efforts they put into reviewing our manuscript. These are highly positive and constructive reviews. Thank you! We have updated our manuscript to include further discussion of several important points (as suggested by reviewers) and addressed reviewer suggestions individually below.

**Reviewer #1 (Public Review):**
This remarkable and creative study from the Asbury lab examines the extent to which mechanical coupling can coordinate the growth of two microtubules attached to isolated kinetochores. The concept of mechanical coupling in kinetochores was proposed in the mid-1990s and makes sense intuitively (as shown in Fig. 1B). But intuitive concepts still need experimental validation, which this study at long last provides. The experiments described in this paper will serve as a foundation for the transition of an intuitive concept into a robust, quantitative, and validated model.The introduction cites at least 5 papers that proposed mechanical coupling in kinetochores, as well as 5 theoretical studies on mechanical coupling within microtubule bundles, so it's clear that this manuscript will be of considerable interest to the field. The intro is very well written (as is the manuscript in general), but I recommend that the authors include a brief review of the variable size of k-fibers across species, to help the reader contextualize the problem.

We agree with the reviewer’s suggestion and have added a brief review of variable k-fiber sizes to the Introduction section (lines 30-35).

For example, budding yeast kinetochores are built around a single microtubule (Winey 1995), so mechanical coupling is not relevant for this species.Indeed, the use of yeast kinetochores to study mechanical coupling is an odd fit, because these structures did not evolve to support such coupling. There is no doubt that yeast kinetochores are useful for demonstrating mechanical coupling and for measuring the stiffnesses necessary to achieve coupling, but I recommend that the authors include a caveat somewhere in the manuscript, perhaps in the place where they discuss their use of simple elastic coupling as compared to viscoelastic coupling or strain-stiffening. It's easy to imagine that kinetochores with large k-fibers might require complex coupling mechanisms, for example.

Even though yeast kinetochores are built around single microtubules, mechanical coupling has still been proposed to help coordinate the dynamics of sister kinetochores in yeast (Gardner et al. 2005, see main text for full reference). We have added this important point to the Introduction section of the manuscript (lines 33-35). The microtubules attached to sister kinetochores are oriented oppositely to one another, in an anti-parallel arrangement that differs from the parallel arrangement we studied here. Nevertheless, it seems likely to us that coordination of anti-parallel microtubule growth between the single microtubules attached to sister kinetochores in yeast relies at least partly on mechanical coupling. One of the many ways we foresee our dual-trap assay being useful in the future is to test how anti-parallel microtubule growth and shortening can be coordinated via mechanical coupling. Of course, since kinetochores can change the dynamics of their attached microtubules (Umbreit et al., 2012, “The Ndc80 kinetochore complex directly modulates microtubule dynamics”), the kinetochores from different species may have also evolved unique mechanisms of modifying microtubule tension-dependent dynamics to achieve coordination of their attached microtubules. Thus far, *in vitro* reconstitutions using kinetochore assemblies from metazoans have not yet achieved the coupling stability that we routinely achieve with isolated yeast kinetochores. As reconstitutions with kinetochores from other species improve, it will be very interesting to test for species-specific differences in how the kinetochores influence microtubule dynamics and in how effectively they can coordinate microtubules via mechanical coupling.

We note that the (visco)elastic properties of yeast kinetochores, and their relative simplicity compared to other kinetochores, shouldn’t significantly affect our primary experimental results. Yeast kinetochores are relatively small and the force on each bead changes very slowly in our experiments (see Figure S3-1 for examples), so the kinetochore’s change in length over time is very slow and very small. We have added this point to the Methods section of the manuscript (lines 479-484). We agree that mechanical coupling in species with large k-fibers might rely on more complex material properties, such as viscoelasticity or strain-stiffening. In principle, that type of complexity could be incorporated into our dual-trap experiments by altering the simulated linker. We view this as an interesting area for future study.

And is mechanical coupling relevant for holocentric kinetochores like those found in *C. elegans*?

This is a very interesting question. While holocentric kinetochores do not form k-fiber bundles (O’Toole et al., 2003, “Morphologically distinct microtubule ends in the mitotic centrosome of Caenorhabditis elegans” and Redemann et al., 2017, “*C. elegans* chromosomes connect to centrosomes by anchoring into the spindle network”), mechanical coupling could be even more important for them compared to monocentric kinetochores because tip-attached microtubules both near each other AND at opposite ends of the same chromosome must grow at similar enough rates to stay attached to the same chromosome. In *C. elegans* prometaphase, opposite chromosome ends move towards the same pole as the chromosome itself oscillates, suggesting that microtubule plus ends attached to the same chromosome are growing in the same direction at the same time (Maddox et al., 2004, ““Holo”er than thou: Chromosome segregation and kinetochore function in *C. elegans*”). Microtubules appear to stop growing or shortening after chromosome alignment is complete (Redemann et al., 2017), at which time the plus ends of kinetochore microtubules are in close proximity to the chromosome surface (O’Toole et al., 2003, Redemann et al., 2017). The tight clustering of kinetochore microtubule tips near the chromosome at metaphase, as well as the coordinated movement of chromosome arms preceding metaphase, suggests a high level of inter-microtubule coordination in the congression leading up to metaphase. We propose this coordination could be achieved by mechanical coupling through the kinetochore proteins on the surface of holocentric chromosomes and through the underlying chromosome itself.

The paper shows considerable rigour in terms of experimental design, statistical analysis, and presentation of results. My only comment on this topic relates to the bandwidth of the dual-trap assay, which I recommend describing in the main text in addition to the methods. For example, the authors note that the stage position is updated at 50 Hz. The authors should clearly explain that this bandwidth is sufficiently fast relative to microtubule growth speeds.

Thank you for this suggestion. We have added to the Results section (lines 131-133) that updating the stage position at 50 Hz is sufficient to maintain the desired force. We also modified the Methods section (lines 488-491) to clarify that the stage position is sampled at 200 Hz, which is more than sufficient to accurately show the growth variability present in dual-trap experiments.

After describing their measurements, the authors use Monte Carlo simulations to show that pauses are essential to a quantitative explanation of their coupling data. Apparently, there is a history of theoretical approaches to coupling, as the introduction cites 5 theoretical studies. I would have appreciated it if the authors had engaged with this literature in the Results section, e.g. by describing which previous study most closely resembles their own and/or comparing and contrasting theirapproach with the previous work.

Thank you for this excellent suggestion. We have added a brief comparison of our work to previous theoretical studies examining the role of mechanical coupling in k-fiber coordination to the Results section (lines 179-185).

Overall, this paper is rigorous, creative, and thought-provoking. The unique experimental approach developed by the Asbury lab shows great promise, and I very much look forward to future iterations.
**Reviewer #2 (Public Review):**
Leeds et al. investigated the role of mechanical coupling in coordinating the growth kinetics of microtubules in kinetochore-fibers (k-fibers). The authors developed a dual optical-trap system to explore how constant load redistributed between a pair of microtubules depending on their growth state coordinates their growth.The main finding of the paper is that the duration and frequency of pausing events during individual microtubule growth are decreased when tension is applied at their tips via kinetochore particles coupled to optically trapped beads. However, the study does not offer any insight into the possible mechanism behind this dependency. For example, it is not clear whether this is a specific property of the kinetochore particles that were used in this experiment, whether it could be attributed to specific proteins in these particles, or if this could potentially be an inherent property of the microtubules themselves.

We agree that the experiments described in our work do not distinguish between tension-dependence inherent to the microtubule itself and tension-dependence conferred by the kinetochore. We speculate about reasons why tension might disfavor pausing in paragraph 5 of the discussion (lines 356-366). Given that microtubule growth is suppressed by compression without the presence of kinetochores or other microtubule-associated proteins (Dogterom & Yurke, 1997, Janson et al., 2003, see main text for full reference), it seems plausible to us that tension-dependent dynamics, including pausing behaviors, might be inherent to microtubules. However, experiments with different tension-bearing plus-end couplers will be required to test this idea rigorously. We view this as an interesting area for future study.

The authors simulate the coordination between two microtubules and show that by using the parameters of pausing and variability in growth rates both measured experimentally they can explain coordination between two microtubules measured in their experiments. This is a convincing result, but k-fibers typically have many more microtubules, and it seems important to understand how the ability to coordinate growth by this mechanism scales with the number of microtubules. It is not obvious whether this mechanism could explain the coordination of more than two microtubules.

We wholeheartedly agree, it is of vital importance to understand how the coordination of growth via mechanical coupling scales with the number of microtubules. Indeed, we have already begun studying simulations of bundles of ten to twenty microtubules based on the pausing model developed in this paper. Simulated microtubule tips appear significantly limited when linked by mechanical couplers of similar stiffnesses to those used in the dual-trap assay, supporting the idea that mechanical coupling may be able to explain much of the coordination between microtubules in growing k-fiber bundles. We hope to use these simulations to continue exploring the degree to which mechanical coupling can coordinate k-fiber microtubules in future publications.

The range of stiffnesses chosen to simulate the microtubule coupling allows linkers to stretch hundreds of nanometers linearly. However, most proteins including those at kinetochore must have finite size and therefore should behave more like worm-like chains rather than linear springs. This means they may appear soft for small elongations, but the force would increase rapidly once the length gets close to the contour length. How this more realistic description of mechanics might affect the conclusions of the work is not clear.

While the worm-like chain is likely a better model for individual linker molecules, deformation of the underlying centromeric chromatin is also likely to be important, with viscoelastic properties that are still poorly understood. Rather than using a complicated (viscoelastic or worm-like-chain-based) model with many unconstrained parameters, we felt a simple model with a single stiffness parameter to characterize the coupling material was a better starting point, allowing a straightforward comparison between stiffer and softer coupling. In future work, simulations could be used to study the effects of strain-stiffening and viscoelasticity and ask if these effects might further improve (or degrade) the efficacy of mechanical coupling for coordinating kinetochore microtubules.

The novel dual-bead assay is interesting. However, it only provides virtual coupling between two otherwise independently growing microtubules. Since the growth of one affects the growth of the other only via software, it is unclear whether the same insight can be gained from the single-bead setup, for example, by moving the bead at a constant speed and monitoring how microtubule growth adjusts to the fixed speed. The advantages of the double-bead setup could have been demonstrated better.

Thank you for your suggestion to clarify the advantages of our dual-trap approach compared to single-trap experiments. We have added a paragraph to the Discussion section (lines 315-327) to explain the following points: In a real k-fiber bundle, each microtubule can dynamically adjust its growth speed to the current force being applied. In the same way, the dual-trap assay allows us to examine how both leading and lagging tips dynamically adjust to the other’s growth speed simultaneously. In addition, in our dual-trap assay each microtubule in the pair is grown at the same time relative to preparing the slide and comes from an identical batch of kinetochore-bead and tubulin-containing growth buffer. Any differences in growth speeds between paired microtubules can be attributed to intrinsic microtubule variability, rather than prep-to-prep or sample-to-sample differences in microtubule dynamics.

**Reviewer #3 (Public Review):**
Leeds et al. employ elegant *in vitro* experiments and sophisticated numerical modeling to investigate the ability of mechanical coupling to coordinate the growth of individual microtubules within microtubule bundles, specifically k-fibers. While individual microtubules naturally polymerize at varying rates, their growth must be tightly regulated to function as a cohesive unit during chromosome segregation. Although this coordination could potentially be achieved biochemically through selective binding of polymerases and depolymerases, the authors demonstrate, using a novel dual laser trap assay, that mechanical coupling alone can also coordinate the growth of *in vitro* microtubule pairs.By reanalyzing recordings of single microtubules growing under constant force (data from their own previous work), the authors investigate the stochastic kinetics of pausing and show that pausing is suppressed by tension. Using a constant shared load, the authors then show that filament growth is tightly coordinated when pairs of microtubules are mechanically coupled by a material with sufficient stiffness. In addition, the authors develop a theoretical model to describe both the natural variability and force dependence of growth, using no freely adjustable parameters. Simulations based on this model, which accounts for stochastic force-dependent pausing and intrinsic variability in microtubule growth rate, fit the dual-trap data well.Overall, this study illuminates the potential of mechanical coupling in coordinating microtubule growth and offers a framework for modeling k-fibers under shared loads. The research exhibits meticulous technical rigor and is presented with exceptional clarity. It provides compelling evidence that a minimal, reconstituted biological system can exhibit complex behavior. As it currently stands, the paper is highly informative and valuable to the field.To provide further clarity regarding the implications of their study, the authors may wish to address the following points in more detail:1. Considering the authors' understanding of the quantitative relationship between forces, microtubule growth, and coordination, is the dual trap assay necessary to demonstrate this coordination? What advantages does the (semi)experimental system offer compared to a purely *in silico* treatment?

Thank you for your suggestion to explain the advantages of our dual-trap approach compared to simulations based on previous recordings of individual microtubules growing under tension. We have added a paragraph about this to the Discussion section (lines 315-327). Previously we knew that a shared load should theoretically tend to coordinate a growing microtubule pair, but we did not know how well, nor did we know the degree of variability that would need to be overcome to achieve coordination. Moreover, there are myriad ways one could model the variability and force dependence in microtubule growth, but not all of them can successfully explain the tip separations we now measure between real microtubule pairs. For instance, our non-pausing model, although entirely derived from force-clamp data, had too much variability and too little coordination between microtubule pairs when we compared simulation results to our dual-trap measurements. Thus, the dual-trap assay allows us to test our assumptions about how variability in microtubule growth arises and how mechanical coupling affects it using real microtubules. Reviewer 2 likewise asked about the advantages of the dual-trap approach relative to single-trap experiments, and we suggest also examining our response to their comment above.

What are the limitations of studying a system comprising only two individual microtubules? How might the presence of crosslinkers, which are typically present *in vivo* between microtubules, influence their behavior in this context?

This is a very interesting question. K-fiber microtubules in many organisms are subject to forces along their lattices from crosslinkers that attach them to each other and to other microtubules outside the k-fiber. Bridging fibers, for example, are pushed apart at the spindle equator by kinesin motors like Eg5, and are thought to maintain tension on k-fiber microtubule tips by sliding them towards the pole (Vukusic et al., 2017, “Microtubule Sliding within the Bridging Fiber Pushes Kinetochore Fibers Apart to Segregate Chromosomes"). Passive crosslinkers can also produce diffusion-like forces that drive microtubules to move relative to one another (although to our knowledge this has only been demonstrated with antiparallel microtubules—see Braun et al., 2017, “Changes in microtubule overlap length regulate kinesin-14-driven microtubule sliding”). Testing how these various lattice-based forces might affect k-fiber coordination is of great interest to us, but it is not easy to envision how it could be done in our dual-trap setup, where the two coupled microtubules only interact through mechanical forces and are biochemically isolated from one another (in separate assay chambers). Perhaps a clever new assay could be devised in the future to study the role of crosslinkers in combination with mechanical coupling on the coordination of growing microtubules in parallel.

How dependent are the results on the chosen segmentation algorithm? Can the distributions of pause and run durations truly be fitted by "simple" Gaussians, as indicated in Figure S5-2? Given the inherent limitations in accurately measuring short durations and the application of threshold durations, it is likely that the first bins in the histograms underestimate events. Cumulative plots could potentially address this issue.

The qualitative trends of tension suppressing pause entrance and promoting pause exit seemed to be insensitive to the choices we made in our segmentation algorithm. We have added a paragraph to the Methods section (lines 558-569) to explain how other choices we tried (a smoothing window of 5 s compared to 2 s and a minimum event duration of 0.01 s compared to 1 s) had only mild effects on the measured force sensitivities but did not affect their signs. This suggests that while imposing a threshold duration almost certainly underestimates the number of shorter events, it does not substantially affect our overall conclusion that tension reduces the rate of pause entry, accelerates pause exit, and speeds assembly during the ‘runs’ between pauses.

For segmenting each position-vs-time record into pause and run intervals, we fit the velocity distribution for each individual recording with a mixture of Gaussians. The distributions from some recordings fit quite well to a sum of Gaussians, while others did not fit as well. However, we found that the exact threshold used to separate runs from pauses (typically between 2 and 4 nm/s) had a surprisingly small effect on what the algorithm differentiated as a pause or a run. The segmentation algorithm and its performance on every record we analyzed can be directly viewed by downloading and running our force-clamp viewer, publicly available at https://doi.org/10.5061/dryad.6djh9w16v.

**Reviewer #2 (Recommendations For The Authors):**
In Figure 3a it would be helpful to see the traces of forces applied to individual microtubules. This would help to understand both, how the force is distributed between individual microtubules depending on their dynamic state and also to see the fluctuations of individual forces.

We completely agree that understanding how force is distributed between microtubules in our dual-trap assay is both interesting and of great value. Although we decided not to include force vs time traces in the main figures, please refer to Figure S3-1, which shows the force-vs-time curves corresponding to the example position-vs-time traces displayed in Figure 3a, plus examples from two additional microtubule pairs.